



# Detection and attribution of aerosol-cloud interactions in large-domain large-eddy simulations with ICON

Montserrat Costa-Surós[1], Odran Sourdeval[2,3], Claudia Acquistapace[1], Holger Baars[4], Cintia Carbajal Henken[5], Christa Genz[2,4], Jonas Hesemann[6], Cristofer Jimenez[4], Marcel König[4], Jan Kretzschmar[2], Nils Madenach[5], Catrin I. Meyer[7], Roland Schrödner[4], Patric Seifert[4], Fabian Senf[4], Matthias Brueck[8], Guido Cioni[8], Jan Frederik Engels[9], Kerstin Fieg[9], Ksenia Gorges[9], Rieke Heinze[8], Pavan Kumar Siligam[9], Ulrike Burkhardt[10], Susanne Crewell[1], Corinna Hoose[6], Axel Seifert[11], Ina Tegen[4], and Johannes Quaas[2]

[1]Universität zu Köln
[2]Universität Leipzig
[3]Université de Lille, CNRS, UMR 8518 – Laboratoire D'Optique Atmospherique
[4]Leibniz Institute for Tropospheric Research
[5]Institute for Space Sciences, Freie Universität Berlin
[6]Karlsruhe Institute of Technology
[7]Jülich Supercomputing Centre, Forschungszentrum Jülich
[8]Max Planck Institute for Meteorology
[9]Deutsches Klimarechenzentrum
[10]Deutsches Zentrum für Luft- und Raumfahrt, Institut für Physik der Atmosphäre
[11]Deutscher Wetterdienst

**Correspondence:** Montserrat Costa Surós (mcostasu@uni-koeln.de)

**Abstract.**

Clouds and aerosols contribute the largest uncertainty to current estimates and interpretations of the Earth's changing energy budget. Here we use a new-generation large-domain large-eddy model, ICON-LEM, to simulate the response of clouds to realistic anthropogenic perturbations in aerosols serving as cloud condensation nuclei (CCN). The novelty compared to previous

5 studies is that (i) the LEM is run in weather prediction mode and with fully interactive land surface over a large domain, and (ii) a large range of data from various sources are used for the detection and attribution. The aerosol perturbation was chosen as peak-aerosol conditions over Europe in 1985, with more than five-fold more sulfate than in 2013. Observational data from various satellite and ground-based remote sensing instruments are used aiming at a detection and attribution of this response. The simulation was run for a selected day (2 May 2013) in which over the selected domain of central Europe a large variety of

10 cloud regimes was present.

It first is demonstrated, using satellite aerosol optical depth retrievals available for both 1985 and 2013, that the aerosol fields for the reference conditions and also for the perturbed ones, as well as the difference between the two, were consistent in the model and the satellite retrievals. In comparison to retrievals from ground-based lidar for 2013, CCN profiles for the reference conditions were consistent with the observations, while the ones for the 1985 conditions were not.





Similarly, detection-and-attribution was successful for droplet number concentrations: the ones simulated for the 2013 conditions were consistent with satellite as well as new ground-based lidar retrievals, while the ones for the 1985 conditions were outside the observational range.

For other cloud quantities, including cloud fraction, liquid water path, cloud-base altitude, and cloud lifetime, the aerosol
response was small compared to their natural variability. Also, large uncertainties in satellite and ground-based observations make the detection-attribution difficult for these quantities. An exception to this is the fact that at large liquid water path, the control simulation matches the observations, while the perturbed one shows too large LWP.

The model simulations allowed to quantify the radiative forcing due to aerosol-cloud interactions, as well as the adjustments to this forcing. The latter were small compared to the variability and showed overall a small positive radiative effect. The
overall effective radiative forcing (ERF) due to aerosol-cloud interactions (ERFaci) in the simulation was dominated thus by the Twomey effect and yielded for this day, region, and aerosol perturbation $-2.6\,\mathrm{W\,m^{-2}}$. Using general circulation models to scale this to a global-mean present-day vs. pre-industrial ERFaci yields a global ERFaci of $-0.8\,\mathrm{W\,m^{-2}}$.

## 1 Introduction

Clouds and aerosols contribute the largest uncertainty to estimates and interpretations of the Earth's changing energy budget
(Boucher et al., 2013). In particular, aerosol-cloud interactions continue to be a challenge for climate models and consequently for climate change predictions (Stevens and Feingold, 2009; Feingold et al., 2016; Seinfeld et al., 2016; Rosenfeld et al., 2019). Changes in aerosol burden have an effect on cloud microphysical properties, with more aerosol leading to more numerous cloud droplets. The resulting change in cloud albedo (Twomey, 1974) implies a radiative forcing due to aerosol-cloud interactions.

Cloud adjustments to aerosol-cloud interactions are manifest as changes in horizontal (cloud fraction) and vertical extent
(manifest as liquid water path, LWP) of cloudiness, with consequent impact on the Earth's radiation budget and, thus, climate. With increased aerosol loading and thus increased drop concentrations, at constant LWP, droplets are smaller. One hypothesis for a subsequent adjustment is that the precipitation rates are reduced, implying that the cloud lifetime increases (Albrecht, 1989), and consequently, LWP and cloud fraction increase. But, at the same time, other adjustment processes, such as responses of the cloud mixing and evaporation (Ackerman et al., 2004) occur that partly act in the opposite direction (Stevens and
Feingold, 2009; Mülmenstädt and Feingold, 2018; Gryspeerdt et al., 2019). Recently, Toll et al. (2019) have found strong observational evidence that aerosols cause a weak average decrease in the amount of water in liquid-phase clouds compared with unpolluted clouds, since the aerosol-induced cloud-water increases and decreases partially cancel each other out. The different adjustments take place at the same time and become more complex with ice and mixed-phase processes. Because different effects can compensate each other (Stevens and Feingold, 2009; Mülmenstädt and Feingold, 2018), it is difficult to
observe isolated cloud effects.

It is nevertheless key to observe how clouds behave as aerosols are perturbed (e.g., Quaas, 2015). Given the large variability of clouds, however, and the plethora of processes involved, it seems best to combine detailed observations with high-resolved modelling (e.g., Mülmenstädt and Feingold, 2018).





Cloud-resolving simulations are a very useful tool to investigate aerosol-cloud interactions, and much of the progress in process-level understanding is from conducting sensitivity studies with such models, and analysing model output in detail (e.g., Ackerman et al., 2004; Khain et al., 2005; Xue and Feingold, 2006; Sandu et al., 2008; Small et al., 2009; Feingold et al., 2010; Fan et al., 2013; Seifert et al., 2015; Gordon et al., 2018). New capabilities now emerge with the possibility to perform cloud-resolving simulations over large domains, with realistic boundary conditions at the land surface and driven by the large scale flow in numerical-weather prediction mode (e.g., Heinze et al., 2017; Miltenberger et al., 2018).

Following this idea, in the present study, mechanisms of aerosol-cloud interactions are analysed and evaluated, using observations, and making use of a set ICOsahedral Non-hydrostatic Large Eddy Model (ICON-LEM) simulations over Germany. The key idea is to assess to which extent the model-simulated aerosol-cloud interaction effects might be detected and attributed in comparison to various observational datasets. This is done in terms of quantification of the impact on cloud macro- and microphysics, precipitation, and radiation, as a response of the modification of CCN concentrations, the small aerosol particles necessary for water vapor to condensate and form cloud droplets. Two CCN input configurations were used in the high-resolved ICON-LEM simulations (156 m horizontal resolution), which were performed over Germany on a selected date (2 May 2013). For the control simulation, CCN concentrations as estimated for 2 May 2013 from a detailed aerosol model were used. For the perturbation, CCN concentrations valid for 1985 were selected. At this time, pollution in Europe was at its peak, about four times higher than presently (Smith et al., 2011, see later for more details). Note that in the present study only modifications to the CCN concentration have been taken into account, and not to ice nucleating particles (INP), neither to scattering, nor to the absorbing aerosol properties (aerosol-radiation interactions, in previous literature referred to as direct and semi-direct aerosol effects). The methodology section provides a description of the model (section 2.1) and the CCN perturbation (section 2.2) as well as the observations (section 2.3) used in this study.

The results have been divided into several sections. In order to evaluate the CCN concentrations, in section 3.1 aerosol optical depth (AOD) and CCN are compared with remote sensing observations from satellite and from the ground. After that, the mean vertical profiles of mass and number concentration from the model simulations are revised in section 3.2 and the differences found between perturbed and control simulations are introduced. In the following sections, these differences are explored in comparison to several kinds of observations: liquid-cloud microphysics are compared to satellite and ground-based remote sensing observations in sections 3.3 and 3.4; and the aerosol effects on precipitation, as well as cloud boundaries and cloud cover, and their perturbation, are analyzed and discussed in sections 3.5 and 3.6, respectively. Details on the sensitivity to the CCN perturbation in different cloud regimes are discussed in section 3.7. The simulated effects on the radiation budget are quantified in section 3.8. The results are summarized and the conclusions are discussed in section 4.

## 2 Methodology

### 2.1 Model and simulation setup

The simulations are run with the ICON model, the atmospheric model jointly developed by the German Meteorological Service (Deutscher Wetterdienst, DWD) and the Max Planck Institute for Meteorology for numerical weather prediction (Zängl et al.,



2015) and climate simulations (Giorgetta et al., 2018), respectively. Within the High Definition Clouds and Precipitation for Climate Prediction (HD(CP)$^2$) project, another configuration of the model was developed to perform large-eddy simulations (Dipankar et al., 2015), whose physics package is used in the present investigation. A detailed model description is provided by Heinze et al. (2017); their study also provides a thorough model evaluation with various observations including those in the

the HD(CP)$^2$ Observational Prototype Experiment (HOPE) (Macke et al., 2017). The ICON-LEM includes parameterizations for land surface processes, sub-grid turbulence (3D Smagorinsky turbulence), cloud microphysical processes, and radiative transfer. A key feature is an advanced two-moment liquid- and ice-phase bulk microphysics scheme (Seifert and Beheng, 2006). The simulation is performed in a limited-area set-up of Germany. It is run in three one-way nested resolutions and the triangular discretisations at each of the three domains in which ICON-LEM is run correspond to 625 m, 312 m, and 156 m

horizontal resolution. This analysis focuses on the results at the highest resolution (156 m).

Due to the large computational cost (1 simulated hour consumes 13 real time hours on 300 nodes, equivalent to about 100.000 EUR per simulated day full economic cost of computing time), only one single day was chosen for the model study. Based on the evaluation results from Heinze et al. (2017), the date of 2 May 2013 has been selected. The key reason is that a wide range of cloud- and precipitation regimes was present at this day (Fig. 1 illustrates the cloud conditions, based on satellite

data). High pressure prevailed over Germany on that date, with low to mid-level convective clouds, that were locally produced. Specifically, at 10 h UTC shallow convection started and finally lead to a peak of deeper precipitation-forming convection at 17 h UTC. In the southern and eastern part of the domain, stronger convection occurred accompanied with thick cloud layers in the afternoon along with a frontal passage. However, the convective clouds were shifted to higher latitudes in the model simulations.

Four simulations were conducted (Table 1). The control (C2R) and perturbed (P2R) simulations were initialized at 0 h UTC on 2 May 2013 from boundary conditions from the European Centre for Medium-Range Weather Forecast (ECMWF) analysis. For analysis, the daytime period between 8 h and 20 h UTC is chosen. The control simulation (C2R) used prescribed CCN distributions from 2013. The sensitivity simulation (P2R) that is analysed here in comparison to the control simulation (C2R) used prescribed CCN distributions from 1985 (see next section).

The extra pair of simulations (C1R and P1R, respectively) was conducted in a setup in which the ICON standard description of cloud optical properties in the radiation scheme was applied. In this approach, cloud optical thickness as input to the radiation is computed on the basis of solely the cloud liquid water mixing ratio, without taking into account variable cloud droplet concentrations. This second pair of simulations with CCN for 2013 and 1985 thus does not account for the radiative forcing due to aerosol-cloud interactions (in previous literature also called first aerosol indirect effect, cloud albedo effect, or

Twomey effect), but only for cloud adjustments (such as the cloud lifetime effect or other rapid responses). In turn, the first pair of simulations (C2R and P2R, respectively) allows to calculate the full effective radiative forcing due to aerosol-cloud interactions, including the Twomey effect. The difference between the two pairs of simulations thus allows to isolate the latter.





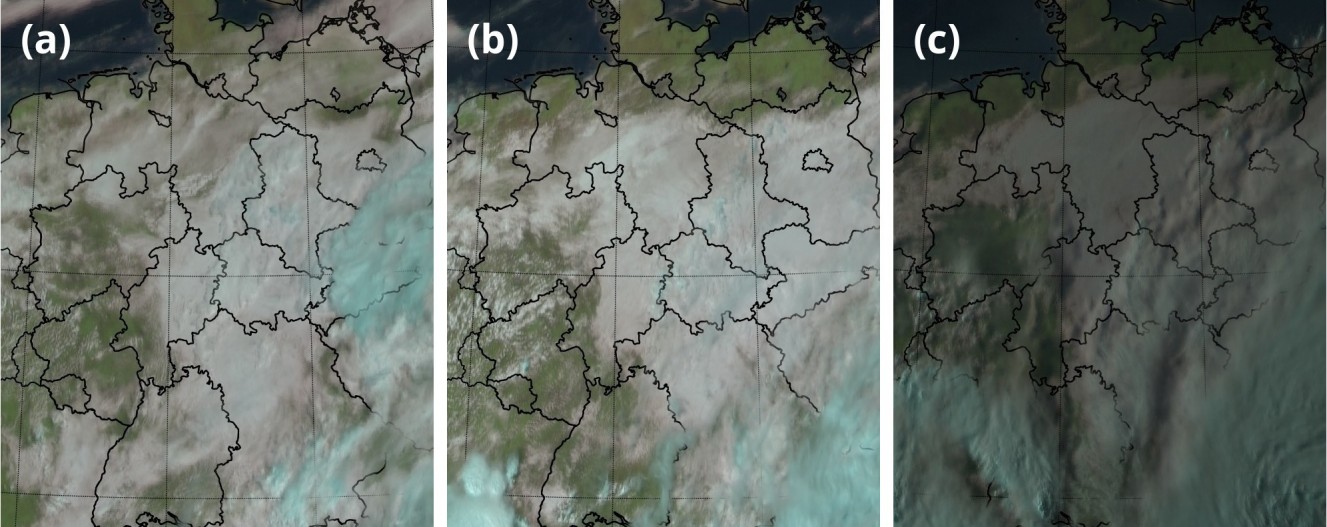

**Figure 1.** Temporal sequence of satellite images at (a) 9 UTC, (b) 13 UTC and (c) 17 UTC on 2 May 2013 as natural color composite of the 0.6-, 0.8-, and 1.6-$\mu$m channels and the high-resolution visible channel from the Spinning Enhanced Visible and Infrared Imager (SEVIRI) instrument on board the geostationary Meteosat satellite. Liquid clouds appear in shades of white and an increase in ice content shifts the color of the clouds towards cyan.

**Table 1.** Summary of simulations for 2 May 2013. Control ("C") runs are with (low) 2013 aerosol concentrations, Perturbed ("P") runs, with (high) 1985 aerosol. The first set takes into account two moments of the cloud particles size distributions in radiation ("2R"), the specific mass and number, whereas the second one, only one moment ("1R"), which is the specific mass.

| Simulation acronym | Microphysics scheme | Clouds in radiation | Aerosol conditions |
|---|---|---|---|
| C2R | 2-moment | Mass and number | 2013 |
| P2R | 2-moment | Mass and number | 1985 |
| C1R | 2-moment | Only mass | 2013 |
| P1R | 2-moment | Only mass | 1985 |

## 2.2 CCN for 2013 and 1985

The anthropogenic aerosol emissions in the Northern Hemisphere exhibited a strong increase since the industrialization starting in the early 18th century and accelerating from the 19th century onwards. Over Europe and North America, they reached a maximum in the 1980s, and declined since then (Smith et al., 2011; Cherian et al., 2014). Over Europe, the peak was at about 5 40 Tg sulfur per year. After that, they decreased to early 1900s levels because of the introduction of air quality policies, and due to economic restructuring in Eastern Europe. Over the course of about 30 years from the mid-1980s (represented here





by the year 1985) and the mid-2010s (represented here by the year 2013), a very substantial change in aerosol concentrations occurred (Smith et al., 2011).

A prerequisite to realistically simulate the aerosol-cloud interactions is a realistic representation of the aerosol concentrations and their capacity to serve as CCN. Compared to the earlier model version (Heinze et al., 2017; Hande et al., 2016), new
time-varying 4D distributions of CCN concentrations were generated from the emissions valid for 2013 and for the peak-aerosol conditions around 1985. The simulation imposing these latter CCN concentrations is hereafter called "perturbed", in comparison to the "control" run which employs CCN concentrations for 2013. The difference between peak-aerosol in 1985 and 2013, rather than, e.g. a comparison of 2013 conditions to pre-industrial aerosol, has been chosen for two reasons: (i) the perturbation is much larger, since the emissions in 2013 were much closer to pre-industrial than to 1985 levels, and (ii) some
observations for 1985 are available to assess the aerosol fields.

The CCN distributions were created with the regional coupled model system COSMO-MUSCAT, which consists of two online-coupled codes: the chemistry transport model Multi-Scale Chemistry Aerosol Transport (MUSCAT) (Wolke et al., 2004, 2012) and the Consortium for Small-scale Modelling (COSMO) model, which is the the operational weather forecast model of the German Meteorological Service (Deutscher Wetterdienst, DWD) and other national meteorological services in Europe
(Baldauf et al., 2011; Schättler et al., 2014). The COSMO model, which is a non-hydrostatic meteorological model and solves the governing equations on the basis of a terrain-following grid (Baldauf et al., 2011), provides MUSCAT with all required meteorological fields. Based on these fields, MUSCAT calculates the transport and transformation processes which include advection, turbulent diffusion, and physico-chemical conversion of particles and trace gases in the air, as well as sink processes (sedimentation, dry and wet deposition) (Knoth and Wolke, 1998; Wolke et al., 2012). The emissions of anthropogenic primary
particles and precursors of secondary aerosols is prescribed using emission fields from the European Monitoring and Evaluation Programme (EMEP, 2009; http://www.emep.int/ ). The emission fluxes of natural primary aerosols (e.g. desert dust, primary marine particles), are computed online within the model depending on meteorological fields (surface wind speed, precipitation) (Heinold et al., 2011).

For the present study, two periods of the year 2013 were simulated with COSMO-MUSCAT, coinciding with the measure-
ment campaigns of the HOPE experiment (Macke et al., 2017). It includes measurements around JOYCE (Löhnert et al., 2015) for the period 26 March – 19 June, and at Melpitz from 1 – 30 September. In order to estimate the aerosol concentrations in 1985, the concentrations of 2013 were scaled using scaling factors for black carbon (BC), sulfate (SU), ammonium sulfate (AS), and ammonium nitrate (AN) (Genz et al., 2019). The scaling factors were derived based on the emission ratios between 2013 and 1985 for the different species. Genz et al. (2019) estimate that the concentration for BC, SU, and AS was larger in
1985 by factors of 2.0, 5.3, and 3.9, respectively, compared to the 2013 concentrations.

Due to the high $SO_2$ emissions in the 1980s, Genz et al. (2019) assumed that there was more than enough sulfate available in order to consume all ammonia to form ammonium sulfate. Therefore, ammonium nitrate is assumed to play a negligible role in the 1980s and consequently its concentration for 1985 was set to zero. Natural aerosol species (sea salt, mineral dust, and organic carbon) are assumed unchanged between the two simulations.





In section 3.1 the AOD derived from the model simulations is compared to satellite AOD retrievals. Since the modeled AOD is calculated at $0.5\,\mu$m, it was scaled to the $0.63\,\mu$m wavelength (taking into account Ångström coefficients for the different species) at which AVHRR retrieves AOD. The calculation of AOD from the modeled results was done following Meier et al. (2012). Moreover, for both HOPE campaigns, the modeled CCN number concentrations are compared against measurements

with a PollyXT lidar systems (Engelmann et al., 2016). In the offline calculation based on COSMO-MUSCAT, the CCN number concentration of the multi-modal size distribution at a fixed supersaturation is calculated according to Abdul-Razzak and Ghan (2000). Details on the used hygroscopicity parameters as well as the derivation of number size distributions from the simulated speciated aerosol mass can be found in Genz et al. (2019).

## 2.3 Observations

### 2.3.1 Satellite-based

AOD as retrieved in the PATMOS-x Advanced Very High Resolution Radiometer (AVHRR) retrievals uses the $0.63\,\mu$m channel. It is only retrieved over sea and in clear-sky cases. The product is a daily average of different NOAA satellites with AVHRR aboard. For 1985 the daily average contains NOAA 7, 8 and 9; and for 2013 data from NOAA 15, 18 and 19 and METOP-B satellite are used.

To compare distributions of liquid cloud properties from the ICON-LEM simulations to satellite observations, the off-line diagnostic tool Cloud Feedback Model Intercomparison Project Observation Simulator Package (COSP; Bodas-Salcedo et al., 2011; Swales et al., 2018) was applied to the ICON simulation output. COSP allows for consistency between cloud properties simulated by ICON and retrieved by satellite observations such as the Moderate Resolution Imaging Spectroradiometer (MODIS). For 2 May 2013, collection-6 cloud products MOD06/MYD06 (Platnick et al., 2015, 2017) from four MODIS satel-

lite overpasses within the domain (MODIS-Aqua at 11.45 h UTC and 13.20 h UTC, MODIS-Terra at 9.55 h UTC and 11.35 h UTC) were analyzed. The ICON-COSP simulations were temporally and spatially matched to the satellite observations as well as regridded to the MODIS data resolution of 1 km. Only cloudy satellite pixels with assigned liquid cloud phase as well as good quality and solar zenith angles below 50° were considered, to exclude uncertain/problematic cloudy retrievals.

For liquid clouds, cloud droplet number concentration ($N_\mathrm{d}$) is derived from cloud effective radius ($r_\mathrm{e}$) and cloud optical

thickness ($\tau_\mathrm{c}$) as in Quaas et al. (2006); where $\alpha = 1.37\cdot10^{-5}\mathrm{m}^{-0.5}$:

$$N_\mathrm{d} = \alpha\,\tau_\mathrm{c}^{0.5}\,r_\mathrm{e}^{-2.5}, \tag{1}$$

an approach that assumes an adiabatic growth of clouds (Grosvenor et al., 2018).

### 2.3.2 Ground-based

A comprehensive set of active and passive remote sensing instruments is part of the Leipzig Aerosol and Cloud Remote Obser-

vations System (LACROS) (Bühl et al., 2013). Specifically, its multiwavelength-Raman-polarization lidar provides backscatter and extinction profiles almost continuously (Baars et al., 2016). In this study it serves to retrieve aerosol profiling variables, and





CCN number concentration, $r_e$, liquid water content ($q_l$), and $N_d$, which are compared with the model output (Section 3.4). The CCN concentration profiles of the lidar measurements were calculated using the method described in Mamouri and Ansmann (2016). Profiles of liquid cloud microphysical properties were derived from ground-based remote sensing using the recently established dual-field-of-view (DFOV) lidar techniques (Grosvenor et al., 2018). Such observations are available at Leipzig,

Germany (51.3°N, 12.4°E), since 2013 and provide information about aerosol-cloud-interaction processes (Schmidt et al., 2014). Originally, the observations were based on the DFOV Raman lidar technique (Schmidt et al., 2013), which can only be applied to nighttime lidar observations in order to reduce effects of the solar background on the measurements of Raman-scattering lidar returns from nitrogen molecules. Progress that was made recently in the accuracy of polarization measurements with lidar (Jimenez et al., 2019a) allows to apply an alternative technique for profiling of liquid-cloud microphysical properties

even during daytime (Jimenez et al., 2017, 2019b). In this novel DFOV-polarization approach, the liquid water content, $q_l$, profile is assumed to increase adiabatically with height from the cloud bottom, while the $N_d$ remains constant. An inversion scheme exploits a non-ambiguous relation between $r_e$ and the extinction coefficient, both dependent on $N_d$ and $q_l$, with the single-FOV depolarization ratio and the relative depolarization, quantities that a DFOV polarization lidar can measure (Jimenez et al., 2017, 2019b).

The cloud radar (8.6 mm wavelength) also based in the MOL-RAO, is well suited for the study of thin, low-reflectivity clouds such as non-drizzling and drizzling stratocumulus clouds, due to its high sensitivity. The cloud radar transmits linear polarized radiation at 35.5 GHz and simultaneously receives the co- and cross-polarized backscattered signal. Observations in zenith mode are used, with an integration time of 1 s and a 256 point Fourier transform for generating the Doppler spectrum. Forward simulations of radar Doppler spectra and their corresponding moments have been performed from the ICON-LEM simula-

tion output using the radar forward simulator included in the Passive and Active Microwave radiative TRAnsfer (PAMTRA) framework (Maahn, 2015). The moments of the synthetic Doppler spectra are derived in the same way as for the observations (Acquistapace et al., 2017), and then the drizzle detection criterion described in Acquistapace et al. (2019) is applied to them. Only liquid clouds are selected following Cloudnet criteria (Illingworth et al., 2007). In section 3.5 the ICON-LEM forward simulated reflectivities have been compared to the radar observations for four different categories of drizzle-cloud droplets-rain

drops.

The ceilometer network of DWD provides backscatter measurements and cloud base height (CBH) retrievals from Jenoptik ceilometers (model CHM15k) at very high temporal resolution (15 s) up to 15 km above ground at 51 stations across Germany (Wiegner and Geiß, 2012; Wiegner et al., 2014; Martucci et al., 2010, http://www.dwd.de/ceilomap) within the ICON-LEM domain. CBH, cloud occurrence as an estimator of cloud cover (CC) (see methodology in Costa-Surós et al., 2013), and cloud

persistences (CP) derived from the ceilometer network are compared with those simulated at the same locations with the ICON-LEM. The methodology to derive the CP is the following: the time resolution from 51 CBH timeseries from ceilometer observations (15 s resolution) was changed to 1 min, which matches the output frequency from ICON-LEM diagnostics. All three 12-hour (8-20 h) timeseries (ceilometer observations, control and perturbed simulations) were splitted into 30 min intervals. For each of these 30 min intervals the CBH between 0 – 3000 m was checked, and if there was a CBH it was flagged as

cloudy and the neighbour cloudy pixels were assigned to one cloud, so a cloud persistence for each single cloud of the interval





was assigned. Then each 30-min interval was classified according to its cloud cover as "clear-sky" (0 – 5 %), "few clouds" (5 – 25 %), "scattered clouds" (25 – 50 %), "broken clouds" (50 – 87 %), or "overcast sky" (87 – 100 %). After that, a normalized histogram was plotted using the cloud persistences from the time intervals classified as "few", "scattered" and "broken" clouds, i.e. cloud cover between 5 – 87 % (see section 3.6).

## 3 Results

### 3.1 Evaluation of the perturbed and unperturbed CCN and aerosol distributions

A comparison of the AOD over the North and Baltic seas as retrieved from AVHRR for 1985 and 2013 and AOD simulated by COSMO-MUSCAT is shown in Fig. 2 and summarized in Table 2. Note that since AVHRR retrieves AOD only in cloudless cases over sea, only a rather small fraction of the time a valid retrieval is available (Table 2). Approximately double AOD levels on average are observed in March–June 1985 in comparison to March–June 2013. The average AOD and also its change between 1985 and 2013 are well captured by the model simulation considering the observation and modelling uncertainty, and natural variability. The geographical distribution over sea (Fig. 2) also shows consistency between the simulation and the satellite retrievals with strong differences between the polluted spring 1985 and the much cleaner spring 2013. Since not very many data-points go into the average from the satellite, the distribution is more noisy compared to the simulation output that is available anytime. Despite this, it is evident that both, observations and model consistently show the expected spatial gradient with larger AOD near the coast and less aerosol over open sea.

**Table 2.** Median aerosol optical depth from AVHRR and COSMO-MUSCAT from 26 March to 19 June in 1985 and 2013. Uncertainty ranges are provided as $25^{th}$ and $75^{th}$ percentiles of regional variability of the temporal mean AOD. In parentheses mean number of days (out of the possible 86 days) per grid box with valid AVHRR retrieval.

|  | 1985 | | 2013 | |
|---|---|---|---|---|
|  | North Sea (9) | Baltic Sea (11) | North Sea (33) | Baltic Sea (30) |
|  | Median [$25^{th} - 75^{th}$] | Median [$25^{th} - 75^{th}$] | Median [$25^{th} - 75^{th}$] | Median [$25^{th} - 75^{th}$] |
| AVHRR | 0.25 [0.22 – 0.30] | 0.30 [0.27 – 0.33] | 0.14 [0.13 – 0.17] | 0.14 [0.13 – 0.15] |
| COSMO-MUSCAT | 0.22 [0.17 – 0.25] | 0.30 [0.27 – 0.32] | 0.09 [0.08 – 0.10] | 0.11 [0.11 – 0.12] |

Profiles of CCN at 0.2 % supersaturation from both COSMO-MUSCAT and from PollyXT lidar retrievals are shown in Fig. 3. The location and time period assessed is the HOPE campaign (Krauthausen, Germany) from 26 March to 19 June 2013. Model and retrieval agree to within the uncertainty in the boundary layer. The mean profiles for 2013 are overestimated by the model by 10–30 % in the boundary layer (from the surface to 1800 m in spring 2013, and 1200 m in fall 2013). Above the boundary layer, the overestimation increases to more than a factor of 2. The estimated CCN concentrations for 1985 are inconsistent with the observations, they exceed the concentration in 2013 by a factor of 2–5 in the boundary layer and up to an





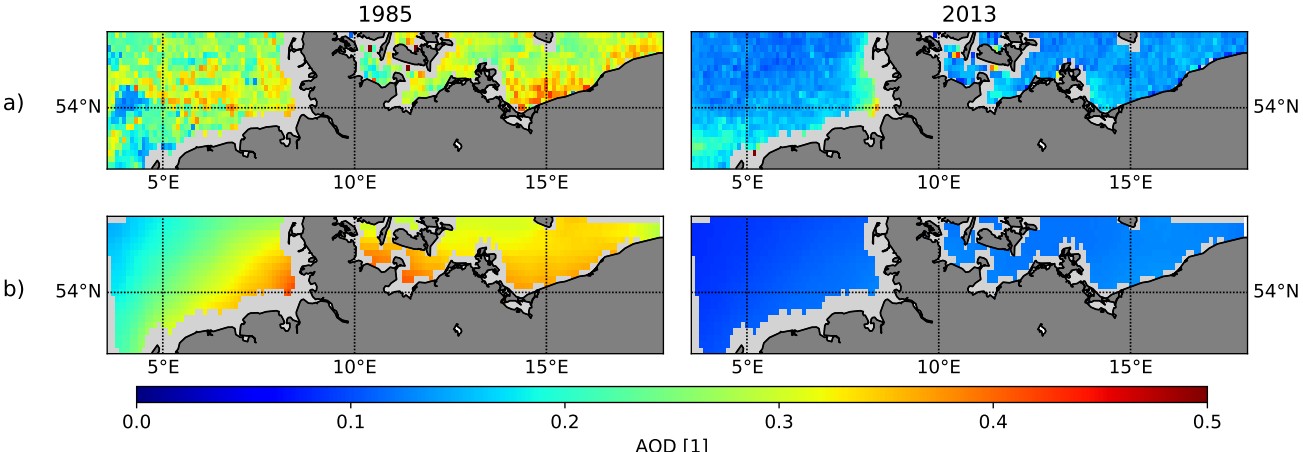

**Figure 2.** Mean AOD retrieved from a) AVHRR (top) and simulated by b) COSMO-MUSCAT (bottom) for 1985 (left column) and 2013 (right). The comparison period covers 26 March to 19 June for each of the years.

order of magnitude in the free troposphere. Similar results are found for the HOPE-Melpitz campaign during September 2013 (not shown).

In conclusion, the imposed aerosol concentrations for 1985 (perturbed simulations) and 2013 (control simulations) match well the distribution and mean values from the satellite retrievals over clear-sky ocean, and it is evident that only the 2013
aerosol yields CCN profiles that are consistent with lidar retrievals from 2013.

## 3.2 Mean vertical profiles of number and mass concentrations: control vs. perturbed simulations

Domain-averaged profiles of number and mass concentration from model output on 2 May 2013 from 8 h to 20 h UTC are shown in Fig. 4. The vertical profile of cloud droplet concentrations closely corresponds to the introduced CCN disturbance (vertical integrated relative increase of 146.9 %, Table 3). In the following sections (3.3 and 3.4) it is explored to which extent
this perturbation is also seen by satellites and from ground-based remote sensing in terms of $N_d$ retrievals. The total-water mass difference at about 2 km altitude (vertically integrated relative increase by 8.8 %) is mainly due to decreased rain in the perturbed simulation (-12.3 %). In the following sections, the changes in the LWP and liquid water content ($q_l$) are investigated.

A slightly higher homogeneous cloud droplet freezing for the perturbed simulation is triggered by upward transport of cloud droplets. In turn, graupel number and mass concentrations are higher in the cleaner environment in low to mid altitudes (3 –
4 km). On the contrary, at higher altitudes (6 – 10 km) the graupel mass is higher in the perturbed simulation.





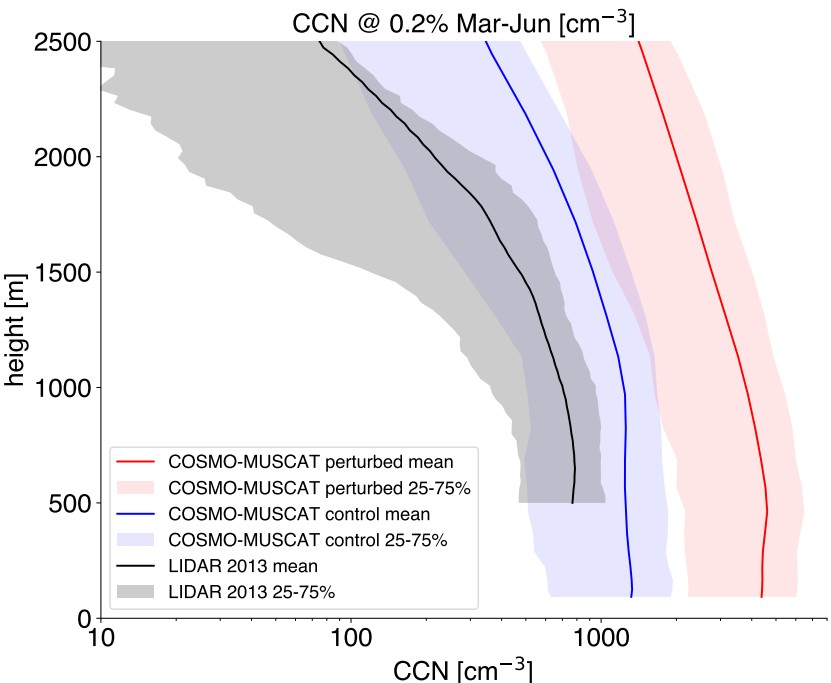

**Figure 3.** Comparison of mean vertical profile of CCN number concentration at 0.2% supersaturation simulated by COSMO-MUSCAT (blue: 2013, red: 1985 aerosol conditions) and retrieved from the PollyXT Lidar measurements using the algorithm of Mamouri and Ansmann (2016) (black), from 26 March to 19 June 2013 during HOPE campaign in Krauthausen, Germany. The shaded area depicts the 25–75 percentiles of the set of single profiles included in the average. A factor 2 of uncertainty is considered for the lidar retrievals, as outlined in Mamouri and Ansmann (2016) and Ansmann et al. (2019).

### 3.3 Liquid-cloud microphysics in comparison to satellite data

Distributions of liquid water path (LWP) and cloud droplet number concentration ($N_d$) from the reference (C2R) and perturbed (P2R) ICON-LEM simulations as well as the corresponding satellite retrievals (MODIS) are shown in Fig. 5. The maximum peak of occurrence and 50% percentile values for $N_d$ compare well between the ICON control simulation (C2R) and MODIS,

5 although the distribution of $N_d$ simulated by ICON-LEM is much broader, resulting in lower and higher 25% and 75% percentile values, respectively, compared to MODIS (Table 4). A reason could be the MODIS instrument sensitivity, since optically very thin clouds are not observed or give problematic retrievals (Grosvenor et al., 2018), and for optically very thick clouds the measurements can go into saturation (note that $N_d$ is computed from the cloud optical thickness and cloud effective radius, see Eq. 1). The simulated $N_d$ distribution for the perturbed simulation (P2R) is shifted to significantly higher values. A factor

10 of about 2 between ICON-control and ICON-perturbed is reflected in the percentile values.





**Table 3.** All-sky domain mean vertically integrated changes between the perturbed and control simulations (P2R – C2R) for number and mass concentrations for water species and for the total water as temporal average from 8 h UTC to 20 h UTC. The absolute changes are given along with the temporal standard deviation of the domain-mean changes. The numbers in brackets are the relative changes.

| Variable | Absolute (relative) all-sky vertically integrated domain-mean day-mean differences (P2R – C2R) | |
| --- | --- | --- |
| | Number concentration [m$^{-2}$] | Mass concentration [kg m$^{-2}$] |
| Total water | $6.9 \cdot 10^{10} \pm 1.3 \cdot 10^{10}$ (147 %) | $3.8 \cdot 10^{-3} \pm 7.7 \cdot 10^{-3}$ (0.9 %) |
| Cloud water | $6.9 \cdot 10^{10} \pm 1.3 \cdot 10^{10}$ (147 %) | $7.6 \cdot 10^{-3} \pm 4.4 \cdot 10^{-3}$ (8.8 %) |
| Cloud ice | $9.7 \cdot 10^{5} \pm 1.0 \cdot 10^{7}$ (0.4 %) | $2.8 \cdot 10^{-4} \pm 7.1 \cdot 10^{-4}$ (0.9 %) |
| Rain | $-3.7 \cdot 10^{6} \pm 1.5 \cdot 10^{6}$ (-25.5 %) | $-5.2 \cdot 10^{-3} \pm 2.5 \cdot 10^{-3}$ (-12.3 %) |
| Snow | $2.7 \cdot 10^{4} \pm 5.6 \cdot 10^{4}$ (2.0 %) | $1.7 \cdot 10^{-4} \pm 5.5 \cdot 10^{-4}$ (1.2 %) |
| Graupel | $-379.3 \pm 1.62 \cdot 10^{4}$ (-1.3 %) | $9.1 \cdot 10^{-4} \pm 6.5 \cdot 10^{-3}$ (0.1 %) |
| Hail | $-1.9 \pm 32.4$ (0.1 %) | $-3.1 \cdot 10^{-3} \pm 3.1 \cdot 10^{-4}$ (0.6 %) |

For LWP values larger than about 10 g m$^{-2}$, MODIS and ICON compare well, both showing occurrence peak values between 100 and 200 g m$^{-2}$. However, ICON (control and perturbed) has higher frequency of low LWP values, resulting in lower 25%, 50%, and 75% percentile values. The model also does not show the bi-modal distribution as MODIS. This bi-modal distribution is due to having two different cloudy scenes in the different overpasses, which happen at different times of the day, i.e. a

cloudy scene with optically thinner clouds and thus lower LWP, and then scenes with optically thicker clouds and higher LWP. However, even if the ICON-LEM output is sampled along the MODIS swath, it does not show this distinct behaviour, but a smooth PDF. The difference between LWP from the ICON reference simulation (C2R) and ICON perturbed simulation (P2R) is small compared to the LWP variability, and small in comparison to the model bias with respect to the MODIS retrievals. It is nevertheless a systematic increase in LWP that is simulated, even if it is small as also expected from recent investigations

of satellite data (Malavelle et al., 2017; Toll et al., 2017; Gryspeerdt et al., 2019; Toll et al., 2019). An exception is at large LWP (larger than 200 g m$^{-2}$), where the control simulation is much smaller than the perturbed one, and closer to the satellite retrievals. This is firstly consistent with the expectation that and increase in $N_{\mathrm{d}}$ leads to an invigoration of convective clouds and, hence, deeper clouds with higher LWP in the tail of the LWP-distribution, where the convective cloud cores can be found. It, secondly, reflects the fact that the thick, precipitating clouds most strongly respond to precipitation delay in response to the

CCN perturbation.

In conclusion, the influence of the perturbed aerosol on $N_{\mathrm{d}}$ clearly can be detected and attributed in comparison to satellite retrievals, but the systematic increase in LWP is too small in comparison to natural variability and model bias to be detected and attributed, except for large LWP ($> 200$ g m$^{-2}$).





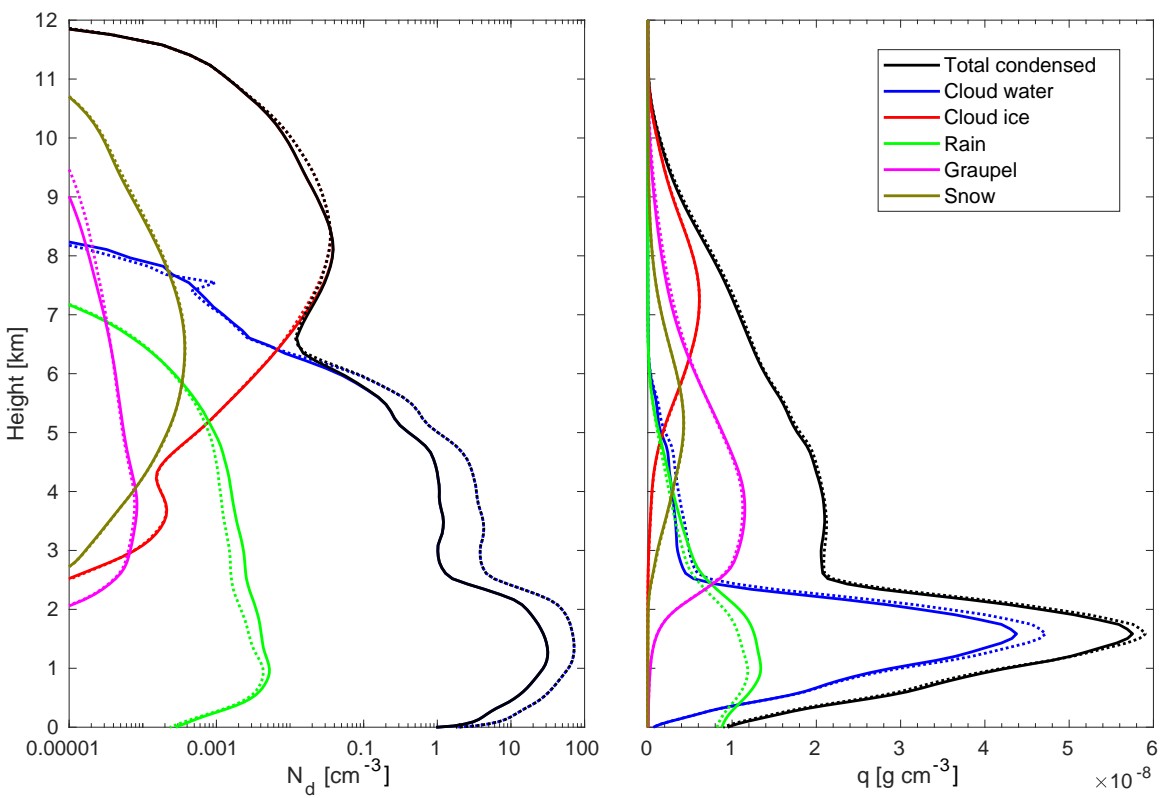

**Figure 4.** Vertical profile of (all-sky) domain mean number concentration (left) and mass concentration (right) of total water and individual particle species on 2 May 2013 from 8 h to 20 h UTC. The control (C2R) simulation is plotted as solid lines and the perturbed (P2R) simulation as dotted lines. Note the logarithmic x-axis for the number concentration panel. Hail not shown due to very low values.

**Table 4.** Median, $25^{th}$ and $75^{th}$ percentiles of liquid water path (LWP) and cloud droplet number concentration ($N_d$) from MODIS and ICON-LEM distributions.

| | $N_d$ [cm$^{-3}$] | LWP [g m$^{-2}$] |
|---|---|---|
| | Median [$25^{th} - 75^{th}$] | Median [$25^{th} - 75^{th}$] |
| MODIS | 170 [97 − 283] | 112 [35 − 203] |
| ICON-LEM-COSP control | 188 [64 − 416] | 68 [23 − 158] |
| ICON-LEM-COSP perturbed | 339 [113 − 808] | 70 [24 − 165] |





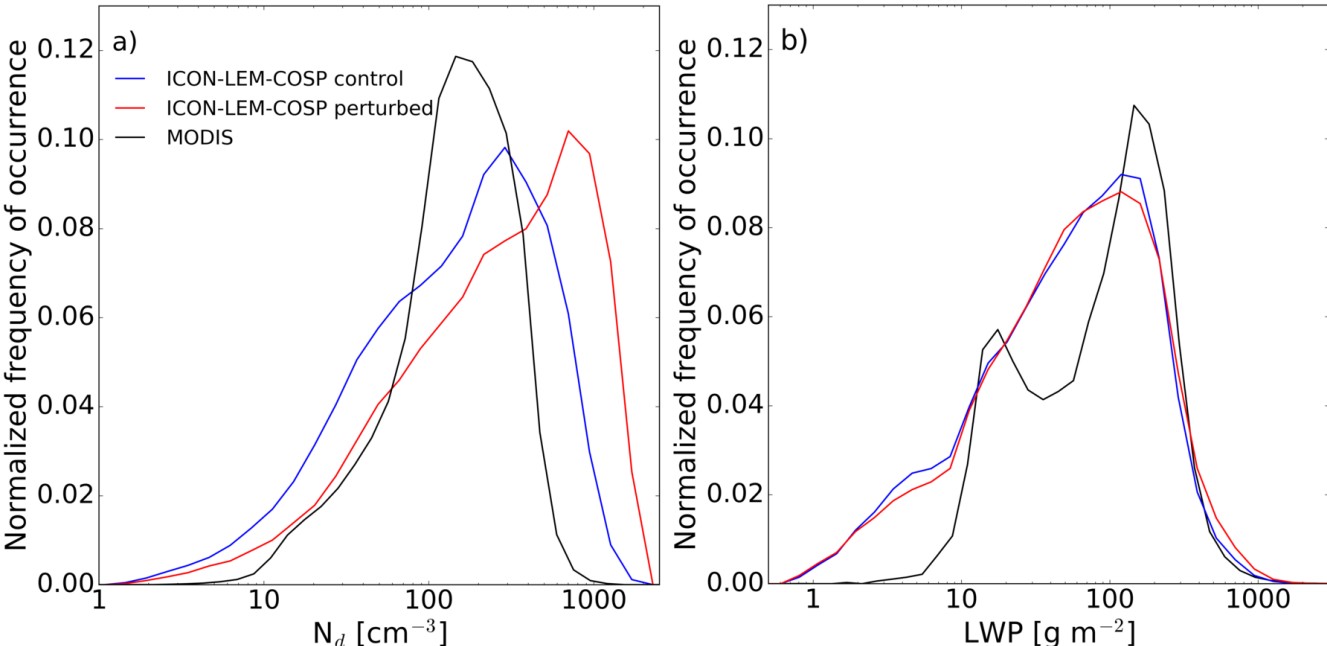

**Figure 5.** Normalized frequency of occurrence distributions computed from ICON-LEM-COSP data from the 2 May 2013 control (C2R, blue) and perturbed (P2R, red) simulations, matched with satellite observations from MODIS (black) obtained at the four satellite overpass times on the same day for a) $N_{\mathrm{d}}$ and b) LWP.

## 3.4 Liquid-cloud microphysics in comparison to ground-based remote sensing

Profiles of several cloud microphysical variables ($N_{\mathrm{d}}$, $r_e$, and $q_{\mathrm{l}}$) were retrieved from ground-based remote sensing as explained in section 2.3.2. In order to derive comparable profiles from ICON-LEM that are best suited for evaluation against the lidar observations, the temporarily high-resolved (9 s) meteogram output was used. All profiles from 21 of the 36 stations for which

meteogram output of ICON-LEM was available (Table A1), were in a first step screened for the occurrence of the presence of hydrometeors. The remaining 15 stations were not considered in the analysis because they were too close (approx. within 20 km) to an already considered station, which would lead to an unwanted weighting of the statistics towards a certain region of the ICON-LEM domain. This was specifically the case for the region of the HOPE campaign (Macke et al., 2017), for which output of 13 stations is available. Vertically continuous sequences of hydrometeors were classified as cloud layers to which

cloud base height and cloud top height were assigned. Each identified cloud layer was in a subsequent processing step filtered in such a way, that (i) precipitation was absent, (ii) ice was absent, (iii) LWP $> 150 \, \mathrm{g \, m^{-2}}$, (iv) cloud depth $< 500 \, \mathrm{m}$, and (v) $1000 \, \mathrm{m} < $ cloud base height $< 4000 \, \mathrm{m}$. Such constraints are similar to the properties of cloud layers which were observed with the DFOV-polarization lidar at Leipzig. The Cloudnet processing suite (Illingworth et al., 2007) operated based on cloud radar, lidar and microwave radiometer observations at Leipzig, was used to identify these conditions for the periods when valid

DFOV-polarization observations were made.





For the evaluation of the ICON-LEM simulations, approx. 40 h of DFOV polarization lidar observations with 3 min temporal resolution (i.e., 800 profiles) were averaged to yield mean vertical profiles of $N_d$, $q_l$, and $r_e$. The cloud periods considered were distributed over 27 days in the spring-summer-autumn seasons of 2017, and over heights between 1 and 4 km above ground level. The resulting profile from the observations and the outputs from the ICON-LEM model for 1985 and 2013 aerosol conditions of 2 May 2013 are presented in Fig. 6. The results corroborate the satellite-based conclusions: for $N_d$, the simulation with the 1985 CCN is inconsistent with the lidar observations, and the one with the 2013 CCN matches the retrievals to within the uncertainty. A similar conclusion is drawn for the effective radius. For $q_l$, in turn – as found before during the comparison to satellite retrieved LWP – the model shows only little aerosol impact, and both realizations compare almost equally to the observations.

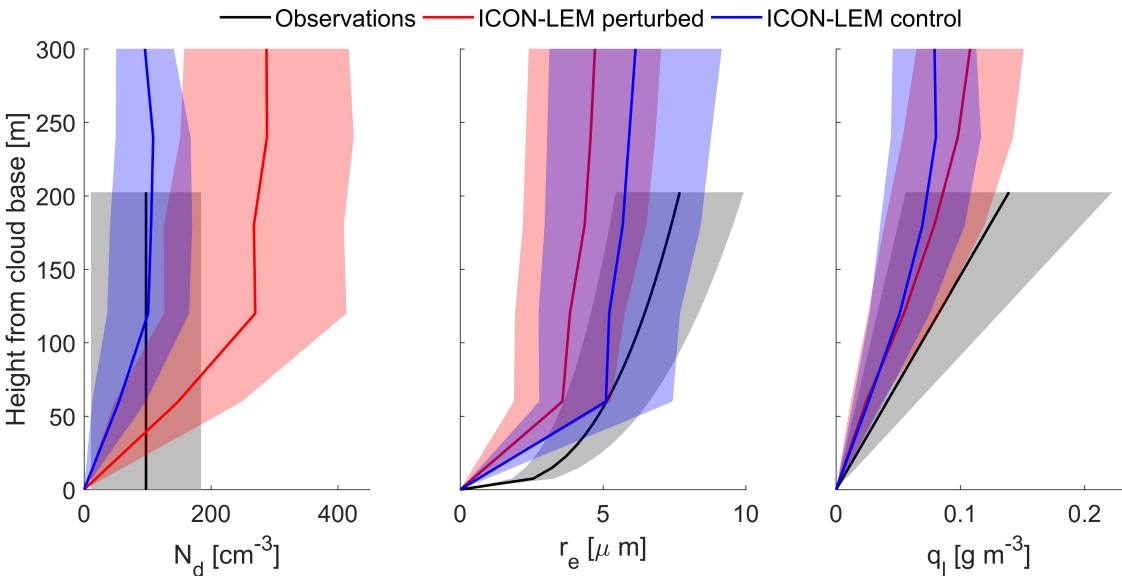

**Figure 6.** Averaged cloud droplet number concentration ($N_d$), cloud droplet effective radius ($r_e$), and liquid water content ($q_l$) profiles retrieved from Lidar (in black, and the temporal variablility as shading in grey) and in ICON-LEM control (in blue) and perturbed (in red) simulations, as a function of height above cloud base.

## 3.5 Effects on precipitation

As discussed earlier, the model simulates large reductions in rain mass and number concentrations, and increases in cloud liquid mass and number concentrations in the perturbed vs. control simulation (Fig. 4). To enquire into the role of the autoconversion process and find out to which extent these changes might be observable, forward simulations of control and perturbed meteogram profiles at the site of the Meteorological Observatory Lindenberg – Richard Aßmann-Observatorium (MOL-RAO) have been performed using the PAMTRA tool (Maahn et al., 2015). After that, the drizzle detection criterion described in Acquistapace et al. (2019) has been applied to the forward simulated Doppler radar moments that were generated with PAMTRA.





This approach is similar to the one presented in Rémillard et al. (2017). An ice cloud mask has been applied to the model output in order to filter out the ice pixels and therefore applying the drizzle detection criterion only to the pixels corresponding to liquid in the cloud mask, in both observations and model output.

The distributions for the different classes of drizzle development are shown in Fig. 7 and summarized in Table 5. The post-5 processed model results shown in Fig. 7 are consistent with the raw results summarized in Fig. 4: the perturbed simulation shows reflectivities that are shifted towards smaller values for all drizzle/precipitation classes, compared to the control simulation. Vice versa, the control simulation produces larger reflectivities, hence larger drops at every stage of drizzle formation, and produces larger raindrops. The simulated values of reflectivities, from both control and perturbed, fall into the range of the observations of MOL-RAO radar, except for the precipitation where both runs overestimate the reflectivities. This overestima-10 tion is also visible from the mean Doppler velocities (proxy of fall speed of hydrometeors) and spectrum width distributions (not shown). Mean Doppler velocities are too high (due to too large drops compared to reality) and this produces also broader spectra, giving larger spectrum widths compared to what observed. A closer look into the radar signal suggests that the small reflectivity values for precipitation observations are due to insects detected by the radar.

**Table 5.** Median as well as 25[th] and 75[th] percentiles of the cloud radar reflectivity (Ze) distributions on 2 May 2013 for the four classes described in Fig. 7.

|  | Drizzle onset Mean [25[th] to 75[th]] | Drizzle growth Mean [25[th] to 75[th]] | Drizzle mature Mean [25[th] to 75[th]] | Precipitation Mean [25[th] to 75[th]] |
|---|---|---|---|---|
| MOL-RAO cloud radar | –18.4 [–22.0 to –15.5] | –24.4 [–30.5 to –17.2] | –13.9 [–22.7 to –7.3] | –40.1 [–51.2 to –30.4] |
| ICON-LEM control (C2R) | –16.2 [–20.5 to –11.7] | –15.6 [–22.3 to –5.2] | –4.9 [–7.0 to –2.3] | –8.1 [–13.5 to –0.9] |
| ICON-LEM perturbed (P2R) | –17.8 [–21.5 to –13.4] | –21.5 [–27.4 to –16.9] | –9.9 [–15.1 to –5.5] | –14.4 [–21.6 to –8.2] |

The comparison between cloud reflectivities (and Doppler velocities and spectrum width) to the corresponding forward 15 modeled variables from ICON-LEM simulations shows a fair agreement between observations and model. However, despite the effort to make model and data comparable, and despite the rather strong signal in the model, this is not yet a useful tool for detection and attribution of differences between control (C2R) and perturbed simulations (P2R) in this case. This tool is at a preliminary stage and can be used to evaluate microphysical schemes to observations (Acquistapace, 2017).

### 3.6 Cloud macrophysics: cloud boundaries, cloud cover and cloud persistence

ICON-LEM cloud base height (CBH), as well as the calculated cloud cover (CC) and cloud persistence (CP) based on the CBH measurements, are compared with high-resolution ceilometer measurements (15 s temporal resolution) from the DWD ceilometer network. In Table 6, mean CBH and CC have been calculated over the 51 stations for 2 May 2013. On average, the model produces less clouds than observed and the cloud base heights are too low in comparison to the ones measured by the ceilometer network. However, the cloud variability is very large, and the model output still is consistent with the data to





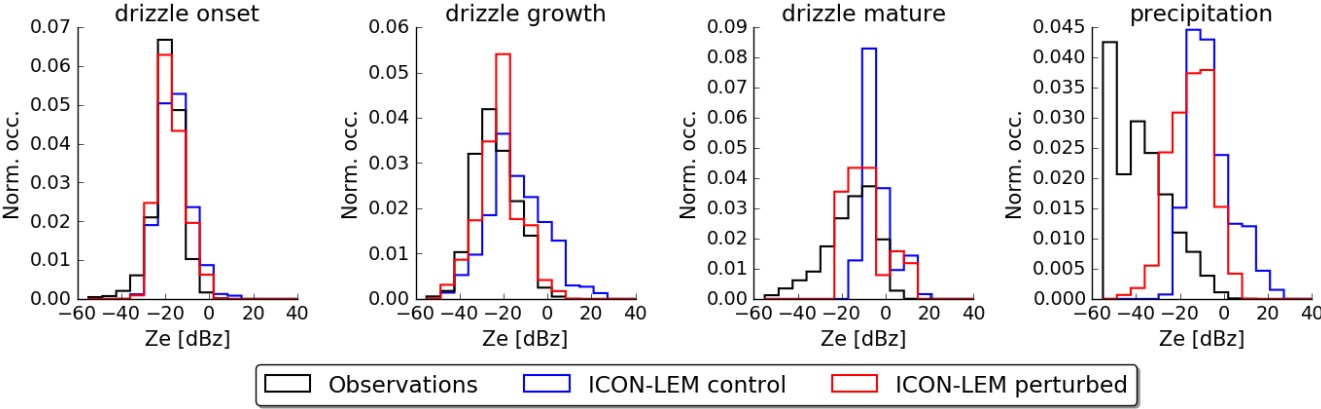

**Figure 7.** Normalized distributions for the different classes of drizzle development in Meteorological Observatory Lindenberg – Richard Aßmann-Observatorium (MOL-RAO) on 2 May 2013. a) drizzle onset, corresponding to the small non-precipitating drizzle drops, larger than cloud droplets but not big enough to fall yet – this is only a signature in the skewness of the Doppler spectrum, not in reflectivity; b) drizzle growth, which contains the drops big enough to modify the spectra shape; c) drizzle mature, which is the drizzle precipitating inside the cloud; and d) precipitation, which is the class of precipitation below cloud base. The radar observations are processed as in Acquistapace et al. (2017), and ICON-LEM meteogram output at the MOL-RAO site is processed using the PAMTRA forward operator and processed analogously. Black: observations; blue: control simulation (C2R); red: perturbed simulation (P2R).

within the uncertainty range. For both variables, the differences between control (C2R) and perturbed (P2R) simulations are so small, in comparison to the observations, that no significant deviations can be detected given the simulation and observation uncertainties. The absolute (relative) differences between perturbed and control simulations are for the mean CBH –4 m (–0.37 %) and for the CC 0.4 % (0.70 %). The same tendency is found in the all-domain simulation means (Table 7) where the

5 perturbed ICON-LEM shows on average higher cloud tops and bases in comparison to the control simulations. The difference between mean cloud top pressures is –263 Pa (0.35 %) and between mean cloud base pressures, –140 Pa (0.17 %). The perturbed simulation also shows higher total cloud cover by 0.16% (relative difference: 0.20 %) in the domain average (Table 7). Cloud fraction can also be assessed from MODIS satellite data and compared to the COSP-processed ICON-LEM output (as in Section 3.3 for LWP and $N_d$). The domain-average cloud fraction for the four MODIS overpasses is 0.84, compared to 0.49 and

10 0.50 for the control and perturbed ICON-LEM simulations, respectively. Despite the very different observational approaches and spatiotemporal sampling, thus, the general conclusion is confirmed: ICON simulates less clouds than observed (a result that also has been noted by Heinze et al., 2017), and shows a positive response of cloud fraction to more aerosol.

Figure 8 shows normalized CBH distributions for the ceilometer network and for both ICON-LEM control (C2R) and perturbed (P2R) simulations, which overestimate very low CBH (< 500 m), and underestimate higher CBH (751 – 1751 m).

Possibly the ceilometer network is not able to detect the latter reliably. Further analysis of 30 min periods exhibit that both simulations overestimate "clear sky" cases (0 - 5 %) and underestimate "overcast" skies (87 - 100 %). "Few" (5 - 25 %) and "scattered" (25 - 50 %) skies are also overestimated, and "broken clouds" (50 - 87 %) are slightly underestimated (not shown).





Cloud persistence analysis for the group of few, scattered, and broken cloud conditions (i.e. cloud cover between 5 % to 87 %) and with bases below 3000 m shows also small differences between control and perturbed distributions (Fig. 8). None of the simulations are able to capture the observed distributions for the short-lived clouds (less than 5 min.), while the longer-lived clouds are overestimated.

**Table 6.** Cloud base height (CBH), cloud persistence (CP), and cloud cover (CC) cover as measured at 51 ceilometer stations in Germany and from ICON-LEM output (2 May 2013 from 8 h to 20 h UTC), as well as CC retrieved from MODIS satellite and ICON-LEM output postprocessed with the COSP simulator. Uncertainty ranges are provided as $25^{th}$ and $75^{th}$ percentile ranges.

| | DWD ceilometer network measurements | | | MODIS |
|---|---|---|---|---|
| | CBH [m] | CP [min] | $CC_{ceilo.}$ [%] | $CC_{MODIS}$ [%] |
| | Mean [$25^{th} - 75^{th}$] | Mean [$25^{th} - 75^{th}$] | Mean [$25^{th} - 75^{th}$] | Mean [$25^{th} - 75^{th}$] |
| Observations | 1454 [774 – 1724] | 4.9 [1.0 – 6.0] | 80.1 [63.9 – 99.5] | 90.4 [88.1 – 92.9] |
| ICON-LEM control (C2R) | 1089 [412 – 1423] | 6.8 [2.5 – 9.0] | 57.1 [32.2 – 84.4] | 52.8 [46.8 – 62.7] |
| ICON-LEM perturbed (P2R) | 1085 [407 – 1428] | 6.7 [3.0 – 9.0] | 57. 5 [32.4 – 83.8] | 53.8 [48.1 – 63.2] |

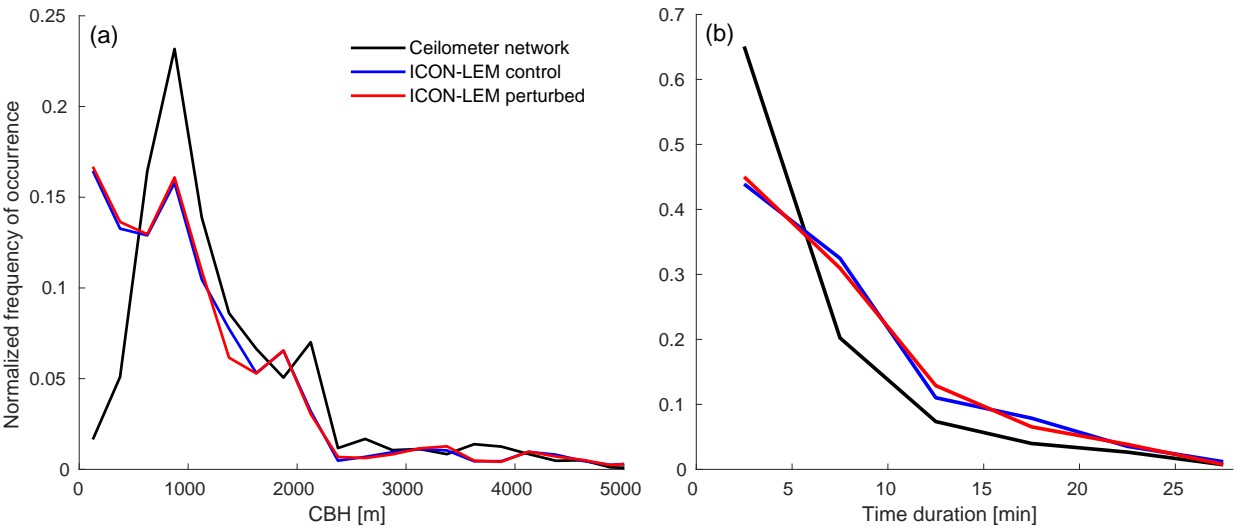

**Figure 8.** Left: Normalized cloud base height distribution (bin size: 250 m) of 51 ceilometer stations over Germany on 2 May 2013 (8 – 20 h UTC). Right: Normalized cloud lifetime (bin size: 5 min) for low-level clouds (<3 km) of few, scattered and broken cloud conditions (i.e. cloud cover between 5 and 87 %). Data from the DWD ceilometer network in Germany (black), and ICON-LEM C2R (blue) and P2R (red) simulations.

5    The comparison of CBH, CC and CP between model and ceilometer measurements shows systematic differences between either model simulation and the observations. Detection and attribution of differences between control (C2R) and perturbed





simulations (P2R) is not feasible in this case. This is mostly because the effect of the aerosol perturbation on these quantities is small compared to the model bias.

## 3.7 Sensitivity to cloud regimes

The International Satellite Cloud Climatology Project (ISCCP) regime classification (Rossow and Schiffer, 1991) utilizes cloud
top pressure and $\tau_c$ to define different cloud regimes. A consistent diagnostics is applied to the ICON output. From the top of the atmosphere down in each column, the first model grid point which has a condensed mass of cloud water plus cloud ice above a threshold of $0.01\,\mathrm{g\,kg^{-1}}$ is utilized to obtain the cloud top pressure (threshold from van den Heever et al., 2010).

An offline computation is used for the total column optical thickness. Cloud top pressure and total column optical depth categorize the clouds into the nine categories, namely: Cumulus, Stratocumulus, Stratus (low level clouds), Altocumulus,
Altostratus, Nimbostratus (mid level clouds), Cirrus, Cirrostratus (high clouds), and deep convective clouds. We applied the classification to the perturbed (P2R) and reference (C2R) model simulation. The classified cloud regime distributions are very similar for the perturbed and control simulations (Fig. 9 shows, as an example, the distribution at 11:45 h UTC). Applying the ISCCP classification to MODIS satellite data yields larger low and mid-level cloud coverage, compared to the model, and less convective cloud coverage. The low and mid-level clouds observed by MODIS are classified as ISCCP-Cirrus and Cirrostratus.
A convective region (clouds with high column optical thickness and high cloud tops) in the south-east is miss-classified by the model, compared to MODIS, while the convective region in the north-east is over-represented in the model. Both Cirrus and convective clouds are most abundant in late afternoon and evening when no MODIS satellite observations were available.

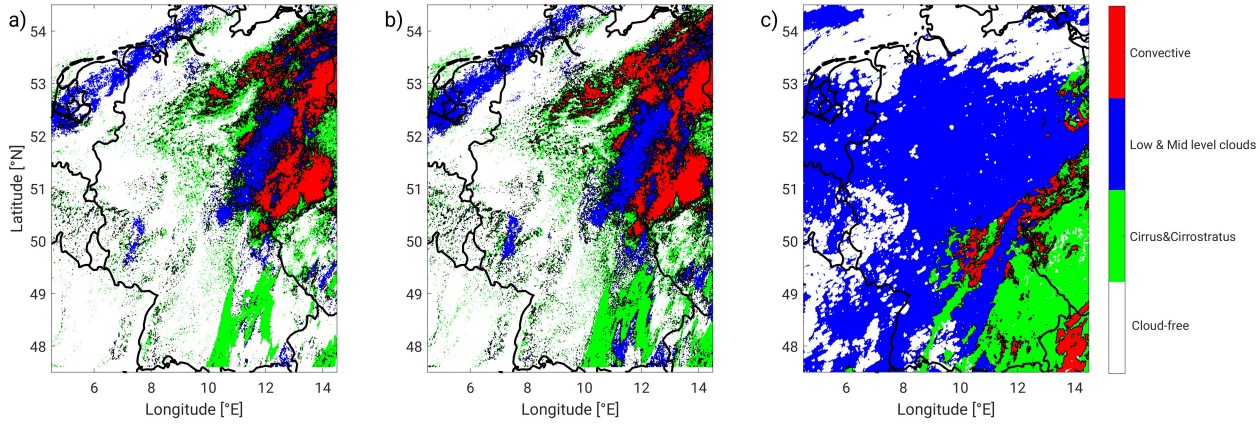

**Figure 9.** Cloudy regimes after application of the ISCCP classifications to ICON-LEM (a) control, (b) perturbed model output, and (c) MODIS at 11:45 h UTC (MODIS overpass time).

Figure 10 displays PDFs of $N_d$, cloud water path, and cloud optical thickness for the entire domain for low and mid-level, convective and high (Cirrus and Cirrostratus) clouds from ICON-COSP simulator output and MODIS retrievals at the four times
of MODIS overpasses (section 2.3). On the one hand, liquid cloud droplet number concentration does show sensitivity for low





and mid-level cloud, since control simulations fits well to MODIS liquid number concentration and perturbed simulation has higher values (already seen in section 3.3). On the other hand, cloud water path is barely sensitive to the CCN perturbation for all cloud regimes considered in the analysis (consistent with the result in section 3.3). However, the control mean values are slightly closer to the observations in all cases. Besides, cloud optical thickness show a notable increase in the perturbed

simulation for low and mid-level clouds, compared to the control simulation. On average, the control simulation is closer to MODIS retrievals than the perturbed simulation. For high and convective clouds, there is not much difference between control and perturbed simulations, and they partly fit to MODIS distribution.

Looking at the different cloud regimes, we can conclude that (1) $N_{\mathrm{d}}$ is a suitable variable for detection and attribution of changes of liquid clouds (low and mid-level clouds), (2) there is a potential use of CWP and COT for detection and attribution

specifically for low and mid-level clouds, (3) the ice concentrations are too similar in the control and perturbed ICON-LEM simulations and so do not allow for an attribution of an aerosol signal of convective and high clouds (Cirrus and Cirrostratus), at least regarding CWP and COT variables.

### 3.8   Radiative implications

The effective radiative forcing due to aerosol-cloud interactions (ERFaci) has been estimated by subtracting the control simula-

tion (C2R) domain averages from the perturbed (P2R) ones (Table 7). For the simulated case, the ERFaci is -2.62±1.80 W m⁻²
in the TOA net solar radiation ($R_{\mathrm{toa}}^{\mathrm{s}}$), and 0.21±0.40 W m⁻² for the TOA net thermal radiation ($R_{\mathrm{toa}}^{\mathrm{t}}$). Consistent with the expectation, the negative forcing in the solar spectrum is slightly reduced by a positive forcing in the terrestrial spectrum (Heyn et al., 2017).

Thanks to the extra simulations that didn't use the number concentration in the radiation transfer computations (C1R and

P1R), which also have two different 4D CCN concentration distributions (for 1985 and 2013), the adjustments to the RFaci, as far as they operate via cloud- and precipitation microphysical and dynamical changes, were quantified. In these simulations only the adjustments associated to aerosol forcing are responsible for the radiation changes. The results are noisy signals: the average changes in cloud fraction and LWP are not different from zero to within the temporal variability (not shown). On average, cloud fraction is simulated to decrease slightly (-0.17 %±0.40 %). This result is surprising and different from what

is seen in satellite statistics (Gryspeerdt et al., 2019). Further analysis is ongoing. The consequence of the decreasing cloud cover, which is more important radiatively than the increase in LWP, is a positive radiative effect of +0.23 ± 1.24 W m⁻².
As difference between the ERFaci and the adjustments, RFaci, or the cloud albedo effect (Twomey, 1974), is obtained as -2.85 W m⁻².

In order to put this number into context, we assessed the aerosol ERF from four different models from the 5th Coupled

Model Intercomparison Project (CMIP5; Taylor et al., 2012) for which the relevant output diagnostics were available to infer the time series of the aerosol ERF. This time series of the ERF over the industrial period was computed by Kretzschmar et al. (2017). The average aerosol ERF (both, aerosol-cloud and aerosol-radiation interactions; the latter are not considered in our current modelling study) for the annual average 1850 to 2000 (present-day minus pre-industrial) as global annual mean was -1.2 W m⁻². The aerosol ERF over central Europe (the domain investigated here) in May, averaged over 1983 to 1987, was



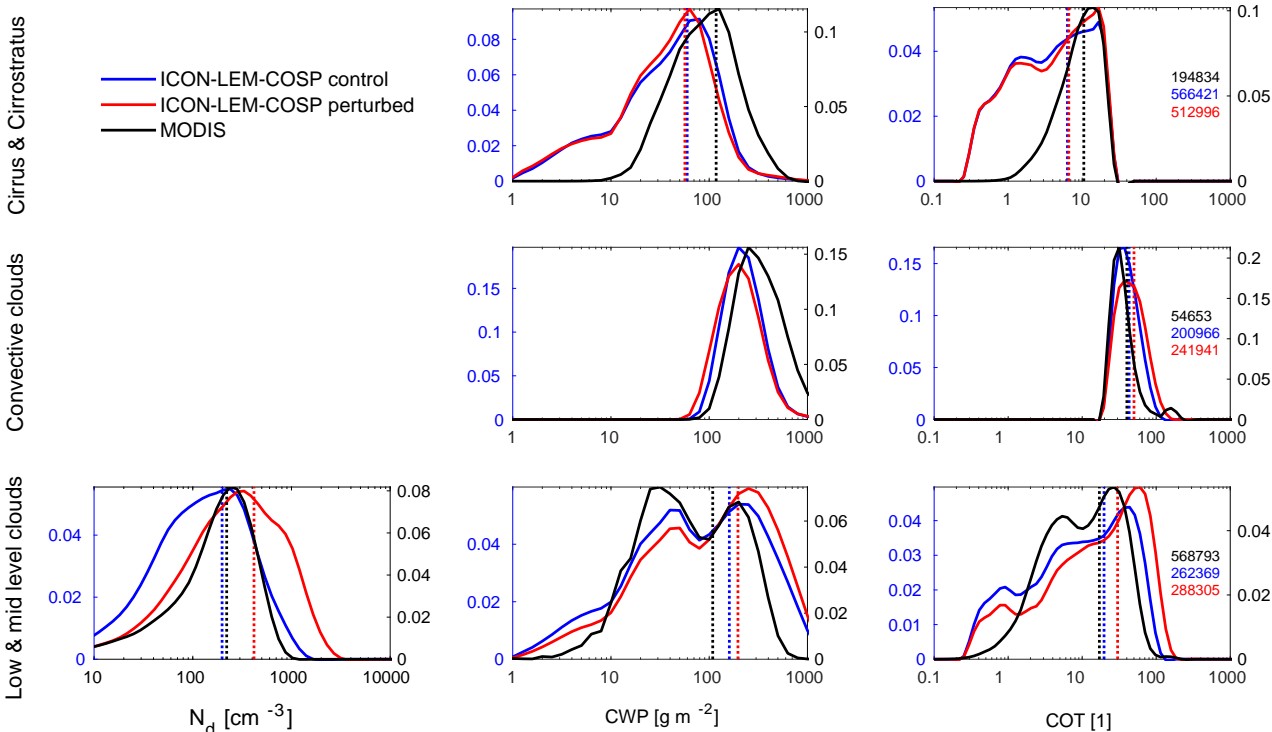

**Figure 10.** Probability distributions of droplet number concentration (left column, only for low, liquid clouds), cloud water path (liquid plus ice water, center column) and cloud optical thickness (right column) of Cirrus and Cirrostratus (top row), convective clouds (second row), and low and mid-level clouds (bottom row) from ICON-COSP simulator and MODIS (in total four overpasses on 2 May 2013). Left y-axis are for ICON-LEM-COSP and right y-axis for MODIS observations. Straight-dotted lines display means. The numbers on the right side show the number of pixels used for the PDF calculation.

-4.0 W m$^{-2}$. This latter is larger than the global-annual average present-day vs. pre-industrial, since it is over a region with a large local aerosol perturbation which is even much larger than present-day minus pre-industrial, and the solar zenith angle in May is larger than in the annual mean. In total, a scaling factor of 3.4 is obtained. The ICON-LEM thus implies a global annual mean ERF due to aerosol-cloud interactions in 2000 of about -0.8 W m$^{-2}$.

## 4 Discussion and Summary

This study used a new type of large-eddy simulation which was carried out over a very large domain and driven as a numerical weather prediction with realistic initial and boundary conditions, including an interactive land surface. A large set of observational data from various sources is used aiming for detection and attribution. Four simulations with ICON-LEM model over





Germany were carried out with different time-varying prescribed CCN concentration distributions. The 4D cloud condensation nuclei concentration inputs generated with COSMO-MUSCAT for 2 May 2013 and 1985 were demonstrated to be consistent with the satellite retrievals of AOD by several AVHRR instruments on different NOAA satellites. Furthermore, the control simulation (C2R) results are consistent with the CCN profile as retrieved from ground-based lidar at two sites in 2013, while

the ones for the perturbed simulation (P2R), with 1985 conditions, are not.

In terms of cloud quantities, it was demonstrated that detection and attribution of the aerosol-induced changes of the droplet number concentration is also possible. The simulated cloud droplet number concentration for the 2013-aerosol simulation (C2R) is consistent with MODIS satellite retrievals, while the perturbed simulation (P2R) results are on average twice higher. An assessment for $N_d$ also was possible from ground-based active remote sensing thanks to a new lidar retrieval technique.

The result confirmed the conclusions on the basis of the satellite data, namely that $N_d$ using the 2013 aerosol is consistent with the 2013 observations, while the perturbed-run output is not.

The other cloud quantities examined included cloud liquid water path, cloud fraction, cloud base height, and cloud lifetime. Satellite data and network ground-based remote sensing were used as observational reference data. For all of these quantities, the ICON model simulated systematic changes between the perturbed (P2R) and control (C2R) aerosol runs. However, in

each case, the difference between either model simulation and the observations was larger than the difference between the simulations in response to the different aerosol conditions. In addition, the natural cloud variability was large compared to the signal. A possible exception is that at large LWP ($> 200\,\mathrm{g\,m^{-2}}$), the control simulation was consistent with the satellite retrievals, while the perturbed simulation showed too large LWP. Small changes between perturbed and control simulations are found in the surface rain rate domain average mean (–2.6 %). However, a detection and attribution even with detailed radar

observations was impossible. The sensitivity analysis of different variables in different cloud regimes may well depend on the synoptic situation. It cannot be excluded that in other synoptic situations the sensitivity in mixed phase clouds and maybe even high clouds may be significant.

The cloud changes lead to an increase in the cloud albedo, with changes in the solar radiation (ERFaci) at the TOA of -2.62 W m⁻². Thanks to a model sensitivity study, the RFaci could be quantified as -2.85 Wm⁻². Using information from global

models, this can be scaled up to the global scale, and the present-day vs. pre-industrial time frame, implying a global ERFaci of -0.8 W m⁻².

**Appendix A**

*Author contributions.*  CCH, PS, UB, SC, CH, IT, AS and JQ conceived the study with input and revisions from all authors. MCS, OS, CA, HB, CCH, CG, JH, CJ, MK, NM, RS, PS, FS performed the analysis with contributions from the other authors. CG, JK, CIM, MB, GC,

JFE, KF, KG, RH and PKS updated the ICON model with the necessary revisions for this study, created necessary input, performed and postprocessed the simulations. MCS, OS and JQ with input from all authors wrote the manuscript.





**Table 7.** Domain mean differences between perturbed and control simulation pair (P2R – C2R). The absolute changes are given along with the temporal standard deviation of the domain mean changes. The numbers in brackets are the relative changes.

| Variable | Absolute (relative) domain mean difference (P2R – C2R) |
|---|---|
| Total cloud cover | $0.16 \pm 0.37\,\%$ $(0.20\,\%)$ |
| Liquid water path | $7.42 \pm 4.09\,\mathrm{g\,m^{-2}}$ $(11.1\,\%)$ |
| Cloud droplet no. conc. ($N_\mathrm{d}$) | $218 \pm 31\,\mathrm{cm^{-3}}$ $(143\,\%)$ |
| Rain rate | $-3.28 \pm 8.46\,\mathrm{g\,m^{-2}\,h^{-1}}$ $(-2.6\,\%)$ |
| TOA net solar radiation | $-2.62 \pm 1.80\,\mathrm{W\,m^{-2}}$ $(-0.58\,\%)$ |
| TOA net terrestrial rad. | $0.21 \pm 0.40\,\mathrm{W\,m^{-2}}$ $(-0.09\,\%)$ |
| Cloud top pressure | $-263 \pm 180\,\mathrm{Pa}$ $(-0.35\,\%)$ |
| Cloud base pressure | $-140 \pm 155\,\mathrm{Pa}$ $(-0.17\,\%)$ |

*Competing interests.* The authors declare that they have no conflict of interests.

*Acknowledgements.* We thank Christine Knist from the Lindenberg Meteorological Observatory – Richard Assmann Observatory for providing us with data from the microwave radiometers; and Björn Stevens for his advice in the "Radiative implications" section. This work is funded by the German Federal Ministry of Education and Research (BMBF) within the framework programme "Research for Sustainable Development (FONA)", www.fona.de, through the research programme "HD(CP)² - High Definition Clouds and Precipitation for Climate Prediction", under the numbers FKZ 01LK1209C, 01LK1212C, 01LK1501E, 01LK1502I, 01LK1503A, 01LK1503E, 01LK1503F, 01LK1503G, 01LK1503H, 01LK1504A and 01LK1507A. The authors gratefully acknowledge the computing time granted through JARA-HPC on the supercomputers JUQUEEN and JURECA at Forschungszentrum Jülich.



**Table A1.** List of stations with ICON-LEM meteogram output used in the scope of this study.

| Station name | Latitude (°N) | Longitude (°E) |
|---|---|---|
| LACROS_HOPE | 50.880 | 6.415 |
| RAO | 52.210 | 14.128 |
| Cabauw | 51.854 | 4.927 |
| ETGB_Bergen | 52.810 | 9.930 |
| EDZE_Essen | 51.400 | 6.960 |
| Greifswald | 54.100 | 13.400 |
| ETGI_Idar-Oberstein | 49.700 | 7.330 |
| ETGK_Kuemmersbruck | 49.430 | 11.900 |
| Meiningen | 50.560 | 10.380 |
| Muenchen-Oberschleissheim | 48.250 | 11.550 |
| Norderney | 53.710 | 7.150 |
| Schleswig | 54.530 | 9.550 |
| Stuttgart | 48.830 | 9.200 |
| Bayreuth | 49.979 | 11.681 |
| Nordholz | 53.778 | 8.668 |
| Ziegendorf | 53.311 | 11.837 |
| Frankfurt_EDDF | 50.035 | 8.555 |
| Duesseldorf_EDDL | 51.288 | 6.769 |
| Hamburg_EDDH | 53.633 | 9.994 |
| Berlin_Tegel_EDDT | 52.560 | 13.288 |
| LACROS_Leipzig | 51.353 | 12.435 |

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
