# Peer review of "Detection and attribution of aerosol-cloud interactions in large-domain large-eddy simulations with ICON"

_Atmospheric Chemistry and Physics, 2019_

## Referee Comment (RC1) · Anonymous Referee #1 · 21 Oct 2019

General comments: The authors present analysis of new very high resolution simulations over the EU domain for one day near peak emissions in 1985 and one day in the present decade with relatively low emissions. The authors carefully analyze the high-resolution simulations using satellite and ground-based data. They find that AOD differences, and Nd differences between 1985 and 2013 are reproduced. Changes in cloud macrophysics are too small relative to natural variability to observe. The authors derive an ERFaci for the global mean using scaling with traditional GCMs.

The paper is a very nice analysis of cutting-edge new simulations and provides an interesting new evaluation of ERFaci. I have the following major issues with the paper:

[Figure]

A lot of the paper is given over to ground-based remote sensing. This is fine, but it is not a field that I am very familiar with and I recommend that a reviewer who is an expert be nominated to comment on this. However, I am concerned by the characterization of standard deviation as uncertainty in comparing observations and models (as discussed in specific comments) and I think this needs to be explained more clearly.

I am not sure that the authors have made a meaningful comment about the adjustment strength, besides the fact that adjustments are small compared to meteorological variability and are hard to see in one day of data- which doesn't preclude them being important to ERFaci.

Critically, I think the scaling to the global ERFaci could be done better (see specific comments below) by expanding the number of GCMs and by showing that the relationship is linear.

Specific comments: Pg1 Ln11: I kind of follow what the authors are trying to say here, but it is a little easy to lose track. I would suggest not using reference and perturbed to refer to 2013 and 1985 in the abstract. It will be easier to follow which conditions are consistent and inconsistent. Effectively it sounds like the model needs the appropriate year of aerosol data to get the right output, which could be said more succinctly.

Pg2 Ln6: What is large LWP? Is it in or area-mean LWP? I am not sure what I should really be taking away from this result. Is it really a key result that needs to be shown in the abstract?

Pg2 Ln16: The results in Rosenfeld 2019 are no longer accurate. There is an errata that revokes most of the findings of the original paper.

Pg3 Ln9 "to what extent"

Pg3 Ln17: Split this into two sentences. CCN is changed and INP is not. Direct effects are not considered. This is really confusing. How is AOD being evaluated if the direct effect isn't considered?

[Figure]

Pg7 Ln17: Please discuss Song et al 2018 (https://www.geosci-model-dev.net/11/3147/2018/gmd-11-3147-2018.html) in the context of using COSP for ICON at this resolution. What is used to drive COSP in this case? Is there subgrid variability assumed?

Pg9 Ln9: Typo- sentence needs to be reordered. Maybe "AOD is only available over the North Sea region for xx% of retrievals." To reduce ambiguity.

Pg9 Ln17: The authors show a systematic difference in the mean CCN profiles from observations and the CCN used to drive the model. I think the authors are somewhat misusing the uncertainty range. Don't you want uncertainty in the mean, not just the variability, which is what this shows? Shouldn't these be standard error in the mean? Ultimately it seems like there is a 10-30% overestimate in CCN relative to the observations (I assume the standard error in the mean is small). Can the authors convert that to an overestimate in Nd using the nucleation scheme, which is the more relevant quantity in this study?

Pg 12 Ln8: I am not sure that 10% change in LWP is small (am I reading table 3 right?). It's certainly true that variability in LWP due to meteorological variability is large, but this doesn't really tell us anything about the radiative forcing induced by adjustments.

Pg12 Ln11 Is the cutoff for large LWP? Does this just mean not thin clouds?

Pg12 Ln18: How many more days of simulation would you need to beat down the noise and be able to see the LWP perturbation clearly?

Pg 20 Ln29 I think the authors are just calculating the ERF over Europe versus the global mean and coming up with a scaling factor. I think a better approach would be to plot ERF_EU_1985-ERF_EU_2013 versus ERF_global_mean_PD for each CMIP5 model. The way that the authors are doing this assumes linearity in this relationship, which is not necessarily true since the EU in 1985 is so polluted. Based on Carslaw et al. (2013), I am not sure that this calculation should really reduce uncertainty much,

but Carslaw et al. (2013) paper implies strong non-linearity in the relationship between local ERF and global-mean ERF. If the authors could increase the number of GCMs beyond 4 and show that the relationship is linear this would be a more robust calculation. How do the authors deal with the direct effect not being calculated in the simulations for this comparison since it will be in the GCMs (Pg3 Ln17)?

Carslaw, K. S., Lee, L. A., Reddington, C. L., Pringle, K. J., Rap, A., Forster, P. M., . . . Pierce, J. R. (2013). Large contribution of natural aerosols to uncertainty in indirect forcing. Nature, 503(7474), 67-71. doi:10.1038/nature12674

———————————————————

---

## Referee Comment (RC2) · Anonymous Referee #2 · 31 Oct 2019

Review of the manuscript numbered ACP-2019-850

Title: "Detection and attribution of aerosol-cloud interactions in large-domain large-eddy simulations with ICON" written by Montserrat Costa-Surós et al. Manuscript number: "acp-2019-850". Decision: "Major revision"

In this study, the authors conducted numerical simulations using large-eddy simulation mode of ICON (ICON-LEM) covering a large calculation domain (whole area of Germany) with fine grid resolution (156 m). They evaluated the ICON-LEM through the comparison between the results of satellite and ground-based observations and those of ICON-LEM. They also tried to detect and attribute the signal of the aerosol effects

on the cloud properties through the sensitivity experiment with changing aerosol. From their analyses, the authors indicated that the signal of the cloud aerosol interaction is only seen in the cloud number concentration and liquid water path larger than 200 g m-2. I think that the nesting simulation using the LES model covering such large domain has never conducted, and this is one of the unique points of this study. This study can be a basis of the numerical weather prediction with such fine grid resolution, and a basis of the numerical studies targeting on aerosol-cloud interaction by "real-case (nesting) simulation" with fine grid resolution. So, I evaluate the authors' efforts to conduct this study. However, most of the analyses conducted in this study can be done by the results of the simulation with "coarse" grid resolution. So, the manuscript has room to be modified as described below. Based on the descriptions shown above, my decision is "major revision", and I encourage the authors to modify the manuscript.

General Comment:

1: As I mentioned before, I evaluate the author's efforts to conduct simulation with fine grid resolution covering such large calculation domain. However, most of the analyses conducted by this study can be done by results of the simulation with coarse grid resolution. The analyses, which can only be done by the results with fine grid resolution, are required. Such analyses extend the value of this study. Entrainment around cloud edge, supersaturation and therefore CCN around the cloud base, and turbulence structure are examples of such analyses (Please do not misunderstand, entrainment, supersaturation, and turbulence are examples).

2: The author concluded that the signal of the aerosol-cloud interaction is difficult to be detected in terms of the cloud cover, cloud top height, cloud bottom height, liquid water path smaller than 200 g m-2. However, is this conclusion applicable for other cases? Based on the previous numerical simulation like Khain et al. (2008), the impacts of the aerosol perturbation on the clouds is dependent upon the meteorological field. I understand that the simulations for other cases using ICON-LEM require huge amount of computational resources, and it is not necessary to conduct the simulations.

However, the author should add comments about whether the conclusion of this study is applicable for other cases or not with referring previous studies.

3: The description about how to couple the aerosol and clouds in the ICON-LEM is not enough. The coupling of the aerosol and cloud is sensitive to the aerosol cloud interaction simulated by the model. In my understanding based on the manuscript, the number concentration of CCN calculated through the results of the COSMO-MUSCAT and the parameterization of Abdul-Razzak and Ghan (2000: AD2000) was given to the microphysical model of Seifeld and Beheng (2006: SB06) in ICON-LEM, and feedback of the cloud to the aerosol field was not calculated like off-line coupling in this study. Is this right? Or is the feedback explicitly calculated? The feedback of the cloud to aerosol (e.g., wet deposition) can reduce the aerosol and CCN number concentration. So, there is a possibility that one of the main conclusions of this study: "signal of the aerosol cloud interaction is limited to the number concentration of clouds (Nd) and LWP larger than 200 g m-2" could be change when the aerosol coupled on-line. Of course, I understand that off-line coupling is good as a first step, but I suggest the authors to add more detailed description about how to couple the aerosol and cloud in ICON-LEM (e.g., how to use CCN number concentration by AD2000 in SB06 with equation).

4: The discussion about the radiative forcing is poor. The authors discussed the radiative forcing for global scale through the scaling of the radiative forcing over the Germany. However, this discussion is unreasonable for the estimation of the global radiative forcing. I think that the discussion about the global radiative forcing is not necessary for this manuscript.

Major Comment:

Line 14 of page 2: Start writing of abstract and introduction are exactly same... I suggest the author to change the start writing of the introduction.

Line 9-10 of Page 4: There are no information about the vertical grid spacing. As well as the horizontal grid spacing, the vertical grid spacing is highly sensitive to the

activation of the cloud around the cloud bottom. The author should add the information about the vertical grid spacing.

Line 10-11 of Page 4: The detail information about the computational resources is not necessary.

Line 12-13: The authors describe the weather condition of target day at this part. The weather map of the target day is helpful for readers to clarify the location of high pressure and frontal system.

Line 15-16 of Page 4: In my understanding, the resolution of ECMWF analysis data is much coarser than ICON-LEM, and it is not suitable for the initial and boundary condition for the simulation with fine grid resolution. The author should be added the detail information of the initial and boundary condition (e.g., resolution, temporal interval, the physical variables used for the initial and boundary condition). In addition, if the initial and boundary condition is much coarser than ICON-LEM model, how do the authors drive the sub-grid scale turbulence? Was the small-scale turbulence, which can be resolved by ICON-LEM but cannot be resolved by ECMWF data, reasonably reproduced after the spin-up time (after 8 hours)?

Line 5-6 of Page 7: As I mentioned in the general comment, the detail descriptions of about how to couple the COSMO-MUSCAT's aerosol and ICOM-LEM are necessary. The detail information about the treatment of the CCN using equations is helpful for readers.

Fig. 2 and Table 2: The AOD simulated with CCN of 2013 is smaller than that observed by satellite. What is the reason of the underestimation of AOD?

Line 21-22 of Page 9: What is the reason of the overestimation of aerosols above the boundary layer? Is the overestimation affects the conclusion of the manuscript? I require the authors to add some comments.

Line 14-15 of Page 10: The authors indicate that graupel number and mass simulated

by clear case are higher at height of 3 – 4 km, but the difference between solid and dotted pink line in Figure 4 is too small to be identified.

Line 2 of Page 11: I think that "Distributions of liquid water path" should be "Probability density frequency (PDF) of liquid water path". Is this right?

Figure 5 and Line 6-7 of Page 11: The authors suggest that the difference in PDF between the model and MODIS is originated from the sensitivity of the MODIS. However, the geographical distribution of cloud simulated by the models are largely different from that observed based on Fig. 9. I think that such difference in the geographical distribution has impacts on the PDF shown in Figure 5.

Line 8-7 of Page 16: "simulated value of reflectivities fall into the range of the observations of MOL-RAO radar" should be "mean simulated value of the reflectivities fall into the range of the observation...".

Line 12-13 of Page 16: The author said the small reflectivity values of for the precipitation observations are due to noise by insects. If the authors know the signal is not originated from the precipitation, the author should remove the noise data.

Line 14-18 page 16: I think this paragraph is not necessary.

Line 20 of Page 16. How did the authors determine the cloud base height and CC simulated by the model? Was this the output of COSP? Usually, the edge of the cloud in the model is determined by a threshold value of LWP or ql. The threshold value is sensitive to the cloud cover and cloud base height. The results in Figure 8 is also sensitive to the threshold value.

Table 6: The authors indicate that the ICON simulate less cloud than observation and CBH is lower than that observed (even though the simulated CBH is included the range of 25-75th of the observation). In my understanding, such difference in the simulated and observed one is usually not originated from the problems in the model used by inner nested domain (i.e. ICON-LEM), but from the data used for initial and boundary

condition (i.e. ECMWF model). So, the author should check the data used for initial and boundary condition or results of outer domain (simulation with the grid spacing of 625 m and 312 m).

Figure 9: As I mentioned in the comment for Figure 5, the difference in the geophysical distribution of simulated cloud and observed one could have some contribution to the difference in PDF shown in Fig. 5.

Section 3.8: As I mentioned in the general comment, the discussion in this part is too rough. Of course, I understand the importance of the estimation of radiative forcing, but the estimation of global averaged ERFaci by the scaling of the results of regional model make readers misunderstanding.

Minor Comment:

Figure 1: The color scale (color bar) is helpful for the readers.

Line 2 of page 5: Reference and detail information of ECMWF analysis data should be added in the list of the reference.

Figure 4 left: For me, it is difficult to identify Black and blue line below the height of 6 km.

Line 6-9 of Page 14: The authors removed the data of the 15 stations because these stations are too close to other stations. I think that the averaged value of the close stations is better for the comparison with the model. The representativeness of the data of selected station is not always confirmed.

Reference:

Khain, A. P., N. BenMoshe, and A. Pokrovsky, 2008: Factors Determining the Impact of Aerosols on Surface Precipitation from Clouds: An Attempt at Classification. J. Atmos. Sci., 65, 1721–1748, https://doi.org/10.1175/2007JAS2515.1.

---

## Author Response (AR1)

*Responses to the interactive comments on "Detection and attribution of aerosol-cloud interactions in large-domain large-eddy simulations with ICON" by Montserrat Costa-Surós et al. to*

**Anonymous Referee #1**

**General comments:** The authors present analysis of new very high resolution simulations over the EU domain for one day near peak emissions in 1985 and one day in the present decade with relatively low emissions. The authors carefully analyze the high-resolution simulations using satellite and ground-based data. They find that AOD differences, and Nd differences between 1985 and 2013 are reproduced. Changes in cloud macrophysics are too small relative to natural variability to observe. The authors derive an ERFaci for the global mean using scaling with traditional GCMs. The paper is a very nice analysis of cutting-edge new simulations and provides an interesting new evaluation of ERFaci.
We thank the reviewer for their thoughtful summary of our study.

I have the following **major issues** with the paper:

A lot of the paper is given over to ground-based remote sensing. This is fine, but it is not a field that I am very familiar with and I recommend that a reviewer who is an expert be nominated to comment on this. However, I am concerned by the characterization of standard deviation as uncertainty in comparing observations and models (as discussed in specific comments) and I think this needs to be explained more clearly. I am not sure that the authors have made a meaningful comment about the adjustment strength, besides the fact that adjustments are small compared to meteorological variability and are hard to see in one day of data- which doesn't preclude them being important to ERFaci. Critically, I think the scaling to the global ERFaci could be done better (see specific comments below) by expanding the number of GCMs and by showing that the relationship is linear.
We thank the reviewer very much for his/her interesting comments. We will proceed to address all of them in the following specific comments.

**Specific comments:**

Pg1 Ln11: I kind of follow what the authors are trying to say here, but it is a little easy to lose track. I would suggest not using reference and perturbed to refer to 2013 and 1985 in the abstract. It will be easier to follow which conditions are consistent and inconsistent. Effectively it sounds like the model needs the appropriate year of aerosol data to get the right output, which could be said more succinctly.
We thank the reviewer for his/her appreciation. The sentence has been simplified in the abstract.

Pg2 Ln6: What is large LWP? Is it in or area-mean LWP? I am not sure what I should really be taking away from this result. Is it really a key result that needs to be shown in the abstract?
We really think this has to be said in the abstract since it is an important result. Following your suggestion, we have added "(LWP >200 g m$^{-2}$)" in the sentence, and clarified it is in-cloud LWP (consistent with the observations).

Pg2 Ln16: The results in Rosenfeld 2019 are no longer accurate. There is an errata that revokes most of the findings of the original paper.
We thank the reviewer for the observation. The reference has been removed from the paper.

Pg3 Ln9 "to what extent".
The typo has been corrected.

Pg3 Ln17: Split this into two sentences. CCN is changed and INP is not. Direct effects are not considered. This is really confusing. How is AOD being evaluated if the direct effect isn't considered?

We thank the reviewer for his/her comment, the sentence has been divided into two and the information extended to be more clear since the direct effect (and the semi-direct) is, in fact, considered but no changes are made to it in the different simulations carried out. That means that the changes in the CCN are not affecting the direct (and semi-direct) effect radiative balance in our simulations. We are able to evaluate the AOD because of the additional offline calculations based on COSMO-MUSCAT, where the CCN number concentration of the multi-modal size distribution at a fixed supersaturation is calculated according to Abdul-Razzak and Ghan (2000), as explained in section 2.2.

Pg7 Ln17: Please discuss Song et al 2018 (https://www.geosci-modeldev.) in the context of using COSP for ICON at this resolution. What is used to drive COSP in this case? Is there subgrid variability assumed?

We thank the reviewer for the Song et al. (2018) reference; this study highlights very well the importance of properly including subgrid variability for GCM evaluation via COSP. However, the conclusions of Song et al. (2018) mainly concern the preferred usage of a model-specific sub-grid information (allowed in COSPv2) instead of the COSP sub-column generator, in order to accurately account for the GCM sub-grid cloud and hydrometeor variability (1.9x2.5deg simulations were used to reach these conclusions). This is not relevant for ICON-LEM, where no sub-grid variabilities of cloud and aerosol properties are considered. Consequently, COSP was used without subcolumn varibility and was driven directly by the grid-level ICON outputs. The goal is here to apply the satellite retrieval algorithm on a pixel level, mainly to reproduce the instrumental sensitivity limitations (i.e. which clouds are too thin to be detected, where does the signal saturate?). Note that the 156-m icosahedral outputs were aggregated into a 1-km^2 lat-lon grid (fitting the MODIS resolution) prior to being used in COSP, but we still decided to not to include sub-grid (< 1km) cloud variability to stay consistent with the MODIS retrieval algorithm, which performs its retrievals by ignoring sub-pixel variabilities.

A new sentence has been added into the "Observation / Satellite-based" subsection: "No sub-column variability is used in COSP, consistently with the lack of sub-pixel variability in MODIS retrievals".

Pg9 Ln9: Typo- sentence needs to be reordered. Maybe "AOD is only available over the North Sea region for xx% of retrievals." To reduce ambiguity.

The sentence has been reformulated.

Pg9 Ln17: The authors show a systematic difference in the mean CCN profiles from observations and the CCN used to drive the model. I think the authors are somewhat misusing the uncertainty range. Don't you want uncertainty in the mean, not just the variability, which is what this shows? Shouldn't these be standard error in the mean? Ultimately it seems like there is a 10-30% overestimate in CCN relative to the observations (I assume the standard error in the mean is small). Can the authors convert that to an overestimate in Nd using the nucleation scheme, which is the more relevant quantity in this study?

We appreciate the reviewer's suggestion. In fact, we chose to show the median and the 25-75$^{th}$ percentiles in purpose because we think they better show the variability of the AOD in the region, rather than the mean and the standard deviation. In this sense, we have removed the word "uncertainty" in Table 2 caption to avoid misunderstandings. Regarding the Nd inquiry, it is discussed in Fig. 3.

Pg 12 Ln8: I am not sure that 10% change in LWP is small (am I reading table 3 right?). It's certainly true that variability in LWP due to meteorological variability is large, but this doesn't really tell us anything about the radiative forcing induced by adjustments.

The reviewer is right, 10 % change is not a small change, however it is too small for detection and attribution of LWP changes by satellite, considering current retrieval uncertainties, therefore the change in LWP is not detectable by MODIS on the studied case, and this is what we mean in the sentence. We clarify that the LWP change translates into a substantial systematic effect on the radiation balance and, thus, the aerosol effective radiative forcing.

Pg12 Ln11 Is the cutoff for large LWP? Does this just mean not thin clouds?

The 200 g m$^{-2}$ value refers to the analysis of Fig. 5. At the large-LWP tail of the PDFs, an increase of high LWP values (higher than about 200 g m-2) clearly appears in the perturbed simulation by comparison to the reference and satellite observations. As explained in this paragraph, we attribute this adjustment effect to invigoration of convective clouds as a consequence of higher Nd. This observation is of particular interest because such adjustments could in principle be detectable based on MODIS-like satellite retrievals. We clarify this in the revised manuscript.

Pg12 Ln18: How many more days of simulation would you need to beat down the noise and be able to see the LWP perturbation clearly?

We could only speculate, since such a method hasn't been applied yet to large-domain large-eddy simulations. In GCM analyses, even for nudged simulations, yearlong integrations are necessary. In an LES we believe a shorter analysis is sufficient due to the very much larger amount of independent columns.

Pg 20 Ln29 I think the authors are just calculating the ERF over Europe versus the global mean and coming up with a scaling factor. I think a better approach would be to plot ERF_EU_1985-ERF_EU_2013 versus ERF_global_mean_PD for each CMIP5 model. The way that the authors are doing this assumes linearity in this relationship, which is not necessarily true since the EU in 1985 is so polluted. Based on Carslaw et al. (2013), I am not sure that this calculation should really reduce uncertainty much, but Carslaw et al. (2013) paper implies strong non-linearity in the relationship between local ERF and global-mean ERF. If the authors could increase the number of GCMs beyond 4 and show that the relationship is linear this would be a more robust calculation.
How do the authors deal with the direct effect not being calculated in the simulations for this comparison since it will be in the GCMs (Pg3 Ln17)?

We thank the reviewer for highlighting that this point requires more attention; reviewer #2 had a very similar concern. The reviewer indeed was right that our previous analysis was overly superficial. Fortunately in the meanwhile, the new 6th Coupled Model Intercomparison Project (CMIP6) provided output from the new multi-model ensemble. This is very valuable to the problem here in question since the part of CMIP6 that addresses the radiative forcing (the RFMIP) has one simulation that allows to diagnose the transient ERF due to aerosols. From this new output, we were now able to assess the scaling in a more thorough way. We explain now in the revised manuscript in much more detail the revised procedure to scaling the forcing, and – more importantly perhaps still in response to this reviewer remark – we much better highlight and quantify the uncertainties. The new approach also allows to better isolate the aerosol-cloud interactions.

Carslaw, K. S., Lee, L. A., Reddington, C. L., Pringle, K. J., Rap, A., Forster, P. M., . . . Pierce, J. R. (2013). Large contribution of natural aerosols to uncertainty in indirect forcing. Nature, 503(7474), 67-71. doi:10.1038/nature12674

*Responses to the interactive comments on "Detection and attribution of aerosol-cloud interactions in large-domain large-eddy simulations with ICON" by Montserrat Costa-Surós et al. to*

**Anonymous Referee #2**

Review of the manuscript numbered ACP-2019-850 Title: "Detection and attribution of aerosol-cloud interactions in large-domain large eddy simulations with ICON" written by Montserrat Costa-Surós et al. Manuscript number: "acp-2019-850". Decision: "Major revision"

In this study, the authors conducted numerical simulations using large-eddy simulation mode of ICON (ICON-LEM) covering a large calculation domain (whole area of Germany) with fine grid resolution (156 m). They evaluated the ICON-LEM through the comparison between the results of satellite and ground-based observations and those of ICON-LEM. They also tried to detect and attribute the signal of the aerosol effects on the cloud properties through the sensitivity experiment with changing aerosol. From their analyses, the authors indicated that the signal of the cloud aerosol interaction is only seen in the cloud number concentration and liquid water path larger than 200 g m-2. I think that the nesting simulation using the LES model covering such large domain has never conducted, and this is one of the unique points of this study. This study can be a basis of the numerical weather prediction with such fine grid resolution, and a basis of the numerical studies targeting on aerosol-cloud interaction by "real-case (nesting) simulation" with fine grid resolution. So, I evaluate the authors' efforts to conduct this study. However, most of the analyses conducted in this study can be done by the results of the simulation with "coarse" grid resolution. So, the manuscript has room to be modified as described below. Based on the descriptions shown above, my decision is "major revision", and I encourage the authors to modify the manuscript.

We thank the reviewer for his/her suggestions. However, we consider that the study could not have been done with the coarser resolution since according to Stevens et al. (2020) there is a clear benefit of using high-resolution simulations (horizontal resolution of 156 m) in comparison to coarser ones (315 m and 625 m) for cloud-related studies.

*Stevens et al., 2020. The Added Value of Large-eddy and Storm-resolving Models for Simulating Clouds and Precipitation, J. Meteor. Soc. Japan., in press (doi: 10.2151/jmsj. 2020-021).*

**General Comment:**

1: As I mentioned before, I evaluate the author's efforts to conduct simulation with fine grid resolution covering such large calculation domain. However, most of the analyses conducted by this study can be done by results of the simulation with coarse grid resolution. The analyses, which can only be done by the results with fine grid resolution, are required. Such analyses extend the value of this study. Entrainment around cloud edge, supersaturation and therefore CCN around the cloud base, and turbulence structure are examples of such analyses (Please do not misunderstand, entrainment, supersaturation, and turbulence are examples).

The reviewer of course is right that similar studies can be performed with coarse-resolution models. Most aerosol-cloud interaction studies so far, in fact, use general circulation models at 1 million times (150 x 150 km² rather than 150 x 150 m²) coarser horizontal resolution. The point here is that the LEM is much better at resolving the relevant cloud processes (cg. Stevens et al., 2020). The high resolution here was also needed for a unique detection-attribution assessment, as it went down nearly to the instrumental scale.

2: The author concluded that the signal of the aerosol-cloud interaction is difficult to be detected in terms of the cloud cover, cloud top height, cloud bottom height, liquid water path smaller than 200 g m-2. However, is this conclusion applicable for other cases? Based on the previous numerical simulation like Khain et al. (2008), the impacts of the aerosol perturbation on the clouds is

dependent upon the meteorological field. I understand that the simulations for other cases using ICON-LEM require huge amount of computational resources, and it is not necessary to conduct the simulations. However, the author should add comments about whether the conclusion of this study is applicable for other cases or not with referring previous studies.

The reviewer is right that it is a clear limitation of our study that only one day over one domain was simulated. However, as stated in Section 2.1 (page 4) the selected date (2 May 2013) covered a wide range of cloud- and precipitation regimes (see Fig. 1, which illustrates the cloud conditions, based on satellite data). The conditions of that day allowed us to study at the same time low, mid, high, and convective clouds, as well as different types of precipitation (see section 3.5). A statement has been added at the end of the Conclusions section that further studies are needed for longer periods and other regions to corroborate, falsify or extend the conclusions.

3: The description about how to couple the aerosol and clouds in the ICON-LEM is not enough. The coupling of the aerosol and cloud is sensitive to the aerosol cloud interaction simulated by the model. In my understanding based on the manuscript, the number concentration of CCN calculated through the results of the COSMO-MUSCAT and the parameterization of Abdul-Razzak and Ghan (2000: AD2000) was given to the microphysical model of Seifeld and Beheng (2006: SB06) in ICON-LEM, and feedback of the cloud to the aerosol field was not calculated like off-line coupling in this study. Is this right? Or is the feedback explicitly calculated? The feedback of the cloud to aerosol (e.g., wet deposition) can reduce the aerosol and CCN number concentration. So, there is a possibility that one of the main conclusions of this study: "signal of the aerosol cloud interaction is limited to the number concentration of clouds (Nd) and LWP larger than 200 g m-2" could be change when the aerosol coupled on-line. Of course, I understand that off-line coupling is good as a first step, but I suggest the authors to add more detailed description about how to couple the aerosol and cloud in ICON-LEM (e.g., how to use CCN number concentration by AD2000 in SB06 with equation).

The reviewer is right in this. We agree that a more detailed description in the text of this work is beneficial for the reader, therefore we added a more extensive statement on this in Section 2.1 to explain that the aerosol is prescribed, but we improved the model by allowing for the CCN sink on activation.

Based on the aerosol species mass modeled by COSMO-MUSCAT, the parameterization described by Abdul-Razzak and Ghan, 2000 is used to calculate time varying 3D fields of the CCN number concentration for a set of updraft velocities. The translation from aerosol mass into aerosol number is done according to Hande et al., 2016, assuming average number size distribution for the different aerosol species. The CCN fields are then used in ICON-LEM.

4: The discussion about the radiative forcing is poor. The authors discussed the radiative forcing for global scale through the scaling of the radiative forcing over the Germany. However, this discussion is unreasonable for the estimation of the global radiative forcing. I think that the discussion about the global radiative forcing is not necessary for this manuscript.

The reviewer raises an important point here that also was raised by reviewer #1. We now substantially expanded the explanations how we obtain the scaling factor. We felt this discussion is important after discussions at a conference where we showed preliminary results: parts of the audience misunderstood the top-of-atmosphere radiation effect over Europe 1985 to 2013 as an aerosol ERF which they compared to the usually-quoted global numbers. This of course is not correct, and so we wanted to help the reader with this short additional computation to understand what the global implications are.

**Major Comment:**

Line 14 of page 2: Start writing of abstract and introduction are exactly same. . . I suggest the author to change the start writing of the introduction.

We thank the reviewer for the observation. The paragraph in the Introduction section has been changed.

Line 9-10 of Page 4: There are no information about the vertical grid spacing. As well as the horizontal grid spacing, the vertical grid spacing is highly sensitive to the activation of the cloud around the cloud bottom. The author should add the information about the vertical grid spacing.
The reviewer is right that this is important information. We added the information about the vertical resolution to the model description.

Line 10-11 of Page 4: The detail information about the computational resources is not necessary.
We agree with the reviewer that is not quite necessary. However, we feel some explanation is required why we only simulate a single day, and to some readers this information maybe useful.

Line 12-13: The authors describe the weather condition of target day at this part. The weather map of the target day is helpful for readers to clarify the location of high pressure and frontal system.
The reviewer is right. We now refer to a former publication (Heinze et al. 2017) for more explanation.

Line 15-16 of Page 4: In my understanding, the resolution of ECMWF analysis data is much coarser than ICON-LEM, and it is not suitable for the initial and boundary condition for the simulation with fine grid resolution. The author should be added the detail information of the initial and boundary condition (e.g., resolution, temporal interval, the physical variables used for the initial and boundary condition). In addition, if the initial and boundary condition is much coarser than ICON-LEM model, how do the authors drive the sub-grid scale turbulence? Was the small-scale turbulence, which can be resolved by ICON-LEM but cannot be resolved by ECMWF data, reasonably reproduced after the spin-up time (after 8 hours)?
The reviewer raises an important point which needed clarification in the text. The text was overly unclear and short on this aspect. We now clarify that indeed it was driven by the COSMO-DE run at 2.8 km and run at three different nests; and refer to the Heinze et al. (2017) paper for more detail.

Line 5-6 of Page 7: As I mentioned in the general comment, the detail descriptions of about how to couple the COSMO-MUSCAT's aerosol and ICOM-LEM are necessary. The detail information about the treatment of the CCN using equations is helpful for readers.
We agree. A more detailed answer is given above as response to general comment #3. We included further information on the calculation of CCN within COSMO-MUSCAT and its usage within ICON-LEM in the revised manuscript.

Fig. 2 and Table 2: The AOD simulated with CCN of 2013 is smaller than that observed by satellite. What is the reason of the underestimation of AOD?
The reason for the deviation between model and observation is not known. In the particular case it is not necessarily an underestimation of the model, but could also be an overestimation by the AOD retrieval. The uncertainty of a single retrieved AOD value is 0.2 (see Zhao et al., 2017), which gives a large relative uncertainty for today's rather clean conditions. If there is a bias after averaging over up to 33 days per pixel is not known. The model naturally also has uncertainties. It seems, that most of the difference occurs near the coast pointing to uncertainties in the exact emissions, e.g. the amount of ships coming (or at least their emissions) out of Hamburg harbour are perhaps underestimated. Most anthropogenic emissions, such as ship tracks, need to be treated on an averaged basis (e.g., monthly or annual average broken down to the integration time step increments).

Zhao, Xuepeng; and NOAA CDR Program (2017): NOAA Climate Data Record (CDR) of AVHRR Daily and Monthly Aerosol Optical Thickness (AOT) over Global Oceans, Version 3.0. doi:10.7289/V5BZ642P.

Line 21-22 of Page 9: What is the reason of the overestimation of aerosols above the boundary layer? Is the overestimation affects the conclusion of the manuscript? I require the authors to add some comments.
Predicting vertical profiles of CCNs with COSMO-MUSCAT is particularly challenging above the planetary boundary layer, and therefore less accurate since the model tends to overestimate the vertical mixing between boundary layer and free troposphere (this information has been added into the manuscript). However, we do not think this affects the conclusion of the manuscript since even if there is an overestimation the values are inside the observations range of uncertainty.

Line 14-15 of Page 10: The authors indicate that graupel number and mass simulated by clear case are higher at height of 3 – 4 km, but the difference between solid and dotted pink line in Figure 4 is too small to be identified.
The reviewer is right that this is indeed not a very clear signal. The sentence has been changed accordingly in the revised manuscript.

Line 2 of Page 11: I think that "Distributions of liquid water path" should be "Probability density frequency (PDF) of liquid water path". Is this right?
We thank the reviewer for his/her suggestion. In this case the area below the curve is not unitary, so it cannot be called "PDF". However, these are normalized frequency of occurrence distributions and can still be interpreted as a kind of probability if one assumes that the distribution is representative for many cases.

Figure 5 and Line 6-7 of Page 11: The authors suggest that the difference in PDF between the model and MODIS is originated from the sensitivity of the MODIS. However, the geographical distribution of cloud simulated by the models are largely different from that observed based on Fig. 9. I think that such difference in the geographical distribution has impacts on the PDF shown in Figure 5.
The reviewer is right, the statement too readily reads as if we blame the discrepancy entirely on the data. Instead, we now first write that the first obvious reason is that the model simulation is far from perfect, but then still remind the reader that also the retrievals are not 100% reliable.

Line 8-7 of Page 16: "simulated value of reflectivities fall into the range of the observations of MOL-RAO radar" should be "mean simulated value of the reflectivities fall into the range of the observation. . .".
We thank the reviewer for his/her correction, the sentence has been changed accordingly.

Line 12-13 of Page 16: The author said the small reflectivity values of for the precipitation observations are due to noise by insects. If the authors know the signal is not originated from the precipitation, the author should remove the noise data.
We appreciate the reviewer's suggestion. Following your comment, we have re-processed the data with the ground clutter removal filtering turned on from 0 to 1600 m AGL.

Here you can compare the two new plots of reflectivity and the old and new figures:

[Figure]

[Figure]

With clutter removal:

[Figure]

The ground clutter removal diminished the height of the precipitation peak of reflectivity (4[th] graph in black) at -60 dbz, however it did not shift the distribution.

By looking at higher spectral moments (see following graphs of mean Ddoppler velocity, Vd, and spectrum width, Sw), one notices that the signal has to be due to clutter, which has not entirely been removed since on that particular day there was a lot of clutter. Observed mean Doppler velocity values are very small, close to zero, indicating non-precipitating targets, typical of ground clutter, and spectrum width values are also very small, around zero. They indicate very narrow spectra shapes, which again are typical of clutter.

[Figure]

The text, the table and the figure in this section has been changed accordingly.

Line 14-18 page 16: I think this paragraph is not necessary.
The reviewer is right, the conclusion is not very specific. We substantially shortened the paragraph to one sentence in the revised version.

Line 20 of Page 16. How did the authors determine the cloud base height and CC simulated by the model? Was this the output of COSP? Usually, the edge of the cloud in the model is determined by a threshold value of LWP or ql. The threshold value is sensitive to the cloud cover and cloud base height. The results in Figure 8 is also sensitive to the threshold value.

We thank the reviewer for pointing out that the text was not clear enough here. As explained in section 2.3.2, the cloud base height is an output from ICON-LEM diagnostics, which is determined as the lowest cloudy grid cell of each column. The threshold for determining a cloudy grid-cell in ICON-LEM is a sum of cloud water and cloud ice ($q_c$ and $q_i$) larger than $10^{-8}$ kg/kg. We now point the reader to this in the revised version.

Table 6: The authors indicate that the ICON simulate less cloud than observation and CBH is lower than that observed (even though the simulated CBH is included the range of 25-75th of the observation). In my understanding, such difference in the simulated and observed one is usually not originated from the problems in the model used by inner nested domain (i.e. ICON-LEM), but from the data used for initial and boundary condition (i.e. ECMWF model). So, the author should check the data used for initial and boundary condition or results of outer domain (simulation with the grid spacing of 625 m and 312 m).

We thank the reviewer for sharing her/his expertise in model skill. In light of this remark, we also checked the data in the other domain resolutions (625 and 312 m) and they give less accurate results than the highest resolution (156 m). We now quote this additional result in the revised manuscript.

Figure 9: As I mentioned in the comment for Figure 5, the difference in the geophysical distribution of simulated cloud and observed one could have some contribution to the difference in PDF shown in Fig. 5.

The reviewer is right with this important remark, and we wrote this in the main text in the revised version.

Section 3.8: As I mentioned in the general comment, the discussion in this part is too rough. Of course, I understand the importance of the estimation of radiative forcing, but the estimation of global averaged ERFaci by the scaling of the results of regional model make readers misunderstanding.

Indeed the reviewer raises a good point that this was too rough. As explained above, without this extra bit, we had the experience that some (other) readers misunderstood the computed effects since they somehow compared the regional, 1985 vs. 2013 results to the global ERFaer they had in mind. So we now considerably added information to this section to make it unambiguous and easier to grasp.

**Minor Comment:**

Figure 1: The color scale (color bar) is helpful for the readers.

We thank the reviewer for his/her suggestion. We fully agree that, in general, a colorbar is very helpful for the reader to understand a shown figure. In our case, we apply a RGB-type mapping in which 3 fields are mapped to a red-green-blue space. This is a very common practice in satellite remote sensing with the advantage that a lot of information can be compressed into one image and it mirrors the way the human eye observes our colored environment. The disadvantage is that no single colorbar can be provided. In our case, we apply a variant of the natural colour RGB (please see https://www.eumetsat.int/website/wcm/idc/idcplg? IdcService=GET_FILE&dDocName=PDF_RGB_QUICK_GUIDE_NCOL&RevisionSelectionMet hod=LatestReleased&Rendition=Web) which combines MSG SEVIRI channels at 0.6, 0.8 and 1.6 micron.

Line 2 of page 5: Reference and detail information of ECMWF analysis data should be added in the list of the reference.

Also in response to the reviewer remark above, we now clarified where the boundary conditions come from and provide the reference to get the additional information in the revised manuscript.

Figure 4 left: For me, it is difficult to identify Black and blue line below the height of 6 km.

We agree with the reviewer. The reason is because the black solid line is over the blue solid one, and the blue and black dashed lines are also one over the other. That means that the most part of contribution to the total condensed water particles come from cloud droplets above 6 km (and from ice particles over 6 km) for both control and perturbed simulations. We have added this information in the figure caption in order to help the reader.

Line 6-9 of Page 14: The authors removed the data of the 15 stations because these stations are too close to other stations. I think that the averaged value of the close stations is better for the comparison with the model. The representativeness of the data of selected station is not always confirmed.

Following the reviewer's suggestion, figure 6 has been updated including all stations available. The following table show the 15 stations that have been added in order to update the statistics for plotting figure 6:

| Station name | Latitude (°N) | Longitude (°E) |
|---|---|---|
| JOYCE | 50.909 | 6.414 |
| KIT_HOPE | 50.897 | 6.464 |
| De_Bilt | 52.100 | 5.180 |
| Lindenberg | 52.210 | 14.110 |
| HOPE-1 | 50.933 | 6.388 |
| HOPE-2 | 50.908 | 6.413 |
| HOPE-3 | 50.899 | 6.459 |
| HOPE-4 | 50.880 | 6.414 |
| HOPE-5 | 50.897 | 6.399 |
| HOPE-6 | 50.864 | 6.425 |
| HOPE-7 | 50.894 | 6.402 |
| HOPE-8 | 50.873 | 6.461 |
| HOPE-9 | 50.925 | 6.423 |
| HOPE | 50.884 | 6.451 |
| Muenchen_EDDM | 48.354 | 11.775 |

Incorporating these additional stations means, two-fold analysis of "De_Bilt", "Lindenberg" and "Muenchen_EDDM" stations (because we were already using a 'twin'-dataset of each station: "Cabauw", "RAO", "Muenchen-Oberschleissheim", correspondingly), and a 12-fold analysis of the HOPE-Jülich area (so far, we were using the gridpoint "LACROS-HOPE").

The following plots show the comparison of the original figure 6 (a) and the updated figure 6 including all stations (b) for the analysis of the liquid clouds:

(a) Selected meteogram stations (as in submitted manuscript)

[Figure]

(b) All meteogram stations (as requested by reviewers)

[Figure]

As the reviewer can see, there's almost no difference between the two figures. Therefore, the manuscript has not been changed in the view of these new results.

*Reference: Khain, A. P., N. BenMoshe, and A. Pokrovsky, 2008: Factors Determining the Impact of Aerosols on Surface Precipitation from Clouds: An Attempt at Classification. J. Atmos. Sci., 65, 1721–1748, https://doi.org/10.1175/.*

**List of changes made in the original manuscript Costa-Surós et al. submitted on ACP**

**Page 1, line 11:**
Original: It first is demonstrated, using satellite aerosol optical depth retrievals available for both 1985 and 2013, that the aerosol fields for the reference conditions and also for the perturbed ones, as well as the difference between the two, were consistent in the model and the satellite retrievals.
Revised: It is first demonstrated using satellite aerosol optical depth retrievals available for both 1985 and 2013, that the aerosol fields for the reference conditions and also for the perturbed ones, as well as the difference between the two, were consistent in the model and the satellite retrievals. used in the model are consistent with corresponding satellite aerosol optical depth retrievals for both 1985 (perturbed) and 2013 (reference) conditions.

**Page 2, line 7:**
Original: An exception to this is the fact that at large liquid water path, the control simulation matches the observations, while the perturbed one shows too large LWP.
Revised: An exception to this is the fact that at large liquid water path (LWP >200 g m−2), the control simulation matches the observations, while the perturbed one shows too large LWP.

**Pag 2, line 14:**
Original: Clouds and aerosols contribute the largest uncertainty to estimates and interpretations of the Earth's changing energy budget (Boucher et al., 2013).
Revised: According to the Fifth Assessment Report of the Intergovernmental Panel on Climate Change (IPCC), clouds and aerosols are largest contributors to uncertainty estimations Cloud and aerosols contribute the largest uncertainty to estimates and interpretations of the Earth's changing energy budget (Boucher et al., 2013).

**Page 2, line 16:**
Original: In particular, aerosol-cloud interactions continue to be a challenge for climate models and consequently for climate change predictions (Stevens and Feingold, 2009; Feingold et al., 2016; Seinfeld et al., 2016; Rosenfeld et al., 2019).
Revised: In particular, aerosol-cloud interactions continue to be a challenge for climate models and consequently for climate change predictions (Stevens and Feingold, 2009; Feingold et al., 2016; Seinfeld et al., 2016; Rosenfeld et al., 2019).

**Page 3, line 12:**
Original: The key idea is to assess to which extent the model-simulated aerosol-cloud interaction effects might be detected and attributed in comparison to various observational dataset.
Revised: The key idea is to assess to what which extent the model-simulated aerosol-cloud interaction effects might be detected and attributed in comparison to various observational dataset.

**Page 3, line 19:**
Original: Note that in the present study only modifications to the CCN concentration have been taken into account, and not to ice nucleating particles (INP), neither to scattering, nor to the absorbing aerosol properties (aerosol-radiation interactions, in previous literature referred to as direct and semi-direct aerosol effects).
Revised: Note that in the present study only modifications to the CCN concentration have been taken into account, and not to ice nucleating particles (INP) neither to. As well, the changes made to the CCN are not affecting the scattering or and nor to the absorbing aerosol properties (aerosol-radiation interactions, in previous literature referred to as direct and semi-direct aerosol effects), but only the aerosol-cloud interactions (in previous literature also called first aerosol indirect effect,

cloud albedo effect, or Twomey effect, and the cloud adjustments, such as the cloud lifetime effect or other rapid responses).

**Page 4, line 7:**
Sentences added: CCN are prescribed in the study (see next Section for more details) as temporally and spatially varying fields read in from offline calculations. The model version used here, however, has been updated to allow for the consumption scavenging of CCN, at droplet activation, a CCN is scavenged. For replenishment, CCN concentrations outside clouds are then relaxed back to the prescribed distribution with a relaxation time scale of 10 minutes.

**Page 4, line 10:**
Sentence added: In the vertical, 150 levels are used, with grid stretching towards the model top at 21 km. The minimal layer thickness is 20 m near the surface and the lowest 1000 m encompass 20 layers.

**Page 4, line 19:**
Sentence added: More details about the weather conditions are given in Heinze et al. (2017).

**Page 4, line 20:**
Original: [...] simulations were initialized at 0 h UTC on 2 May 2013 from boundary conditions from the European Centre for Medium-Range Weather Forecast (ECMWF) analysis."
Revised: […] simulations were initialized at 0 h UTC on 2 May 2013 from  simulations with the COSMO-DE model at 2.8 km resolution, and were run in three nests from coarse (624 m horizontally) to intermediate (312 m) to fine (156 m) resolution as described in Heinze et al. (2017).

**Page 4, line 22:**
Sentence added: In particular, the model output used in the study was on one hand, the so-called meteograms which were temporarily high-resolved (9 s) and available at 36 station locations; and on the other, 2D (at 1 min resolution) and 3D (available every 30 min and at 150 vertical levels) whole domain data fields.

**Page 7, line 5:**
Original: Details on the used hygroscopicity parameters as well as the derivation of number size distributions from the simulated speciated aerosol mass can be found in Genz et al. (2019).
Revised: The calculation of the CCN number concentration (number of activated aerosol particles) follows Hande et al. (2016) and Genz et al. (2019), using the parameterization by Abdul-Razzak and Ghan (2000) for multi-modal size distributions. Details on the used hygroscopicity parameters as well as the derivation of number size distributions from the simulated speciated aerosol mass can be found in Genz et al. (2019). For the comparison to available observations, the CCN number concentration field at a fixed supersaturation is calculated. However, in order to provide CCN fields for ICON-LEM, time varying 3D fields of the CCN number concentration at different constant updraft velocities was required. Abdul-Razzak and Ghan (2000) relate the aerosol composition and updraft velocity to the maximum supersaturation during an air parcels ascend, which in the end determines the number of activated aerosol particles. The calculated CCN fields were then used in the ICON-LEM simulations replacing the fixed assumed CCN number concentration/distribution.

**Page 7, line 24:**
A new sentence has been added: No sub-column variability is used in COSP, consistently with the lack of sub-pixel variability in MODIS retrievals.

**Page 8, line 3:**
A new sentence has been added: for measurements during the HOPE campaign (Section 3.1)

**Page 8, line 6:**
A new sentence has been added: and allow a climatological assessment (Section 3.4)

**Page 8, line 15:**
Original: The cloud radar (8.6 mm wavelength) also based in the (MOL-RAO)
Revised: The cloud radar (8.6 mm wavelength) based in the Meteorological Observatory Lindenberg – Richard Aßmann-Observatorium (MOL-RAO)

**Page 8, line 29:**
Sentence added: The CBH is an output from ICON-LEM diagnostics, which is determined as the lowest cloudy grid cell of each column. The threshold for determining a cloudy grid-cell in ICON-LEM is a sum of cloud water and cloud ice ($q_c$ and $q_i$) larger than $10^{-8}$ kg kg$^{-1}$.

**Page 9, line 12:**
Original: Note that since AVHRR retrieves AOD only in cloudlesscases over sea, only a rather small fraction of the time a valid retrieval is available (Table 2).
Revised: Note that since AVHRR retrieves AOD only in cloudless cases over sea, AOD is only available over the North sea and Baltic sea region for a small fraction of the time (between 10-12 % in 1985, and 35-38 % in 2013,  Table 2).

**Page 9, line 28:**
Original: Above the boundary layer, the overestimation increases to more than a factor of 2.
Revised: Above the boundary layer, the overestimation increases to more than a factor of 2 since the model tends to overestimate the vertical mixing between boundary layer and free troposphere (Heinold et al., 2011a).

**Page 9, Table 2 caption:**
Original: Uncertainty ranges are provided as 25th and 75th percentiles of regional variability of the temporal mean AOD.
Revised:  25th and 75th percentile ranges of regional variability of the temporal mean AOD are provided in brackets.

**Page 10, line 6**
Original: 3.2 Mean vertical profiles of number and mass concentrations [...]
Revised: 3.2 Mean vertical hydrometeor profiles of number and mass concentrations [...]

**Page 10, line 7:**
Original: Domain-averaged profiles of [...]
Revised: Domain-averaged hydrometeor profiles of [...]

**Page 10, line 9:**
Original: relative increase of 146.9 %
Revised: relative increase of 147 %

**Page 11, line 2:**
Original: Distributions of liquid water path […]
Revised: Normalized frequency of occurrence distributions of liquid water path […]

**Page 11, line 4:**
Original: In turn, graupel number and mass concentrations are higher in the cleaner environment in low to mid altitudes (3 – 4 km).
Revised: In turn, graupel number and mass concentrations are slightly higher in the cleaner environment in low to mid altitudes(3 – 4 km).

**Page 11, line 7:**
Original: […] compared to MODIS (Table 4). A reason could be the MODIS instrument sensitivity, since optically […]
Revised: […]  compared to MODIS (Table 4). This can be partly explained by a too large range simulated by the model, which in turn can be partly related to a difference in observed and simulated spatial distribution of clouds at the MODIS observation times. However, a part of the difference in the range of the distributions can be very likely attributed to the MODIS instrument characteristics, since optically […].

**Page 12, line 7:**
Original: […] smooth PDF.
Revised: […] smooth distribution.

**Page 12, line 12:**
Original: An exception is at large LWP (larger than 200 g m$^{-2}$), where […].
Revised: An exception is at large LWP (larger than 200 g m$^{-2}$, see Fig. 5), where […].

**Page 12, line 18:**
Sentence added: The systematic change in LWP, even if small compared to weather variability, implies a substantial contribution to the aerosol effective radiative forcing.

**Page 13, Figure 4 caption:**
Note added: Also note that in the left panel the black solid line is over the blue solid one, and the blue and black dashed lines are also one over the other.

**Page 15, line 14:**
Original: the Meteorological Observatory Lindenberg – Richard Aßmann-Observatorium (MOL-RAO)
Revised: the MOL-RAO

**Page 15, Figure 6 caption:**
Note added: Note that measurements can only provide profiles up to 200 m into the cloud due to the strong extinction of the Lidar signal

**Page 16, line 8:**
Original: The simulated values of reflectivities, [...], fall into the range of the observations of MOL-RAO radar, [...]"
Revised: The mean simulated value of the reflectivities, [...], fall into the range of the observation of MOL-RAO radar, [...].

**Page 16, line 12:**
Original: A closer look into the radar signal suggests that the small reflectivity values for precipitation observations are due to insects detected by the radar.
Revised: A closer look into the radar signal suggests that the small reflectivity values for precipitation observations are due to insects detected by the radar, despite a clutter removal filter was applied to the radar spectra from 0 to 1600 m AGL as pre-processing.

**Page 16, Table 5:**

Original:

**Table 5.** Median as well as $25^{\text{th}}$ and $75^{\text{th}}$ percentiles of the cloud radar reflectivity (Ze) distributions on 2 May 2013 for the four classes described in Fig. 7.

|  | Drizzle onset Mean [$25^{\text{th}}$ to $75^{\text{th}}$] | Drizzle growth Mean [$25^{\text{th}}$ to $75^{\text{th}}$] | Drizzle mature Mean [$25^{\text{th}}$ to $75^{\text{th}}$] | Precipitation Mean [$25^{\text{th}}$ to $75^{\text{th}}$] |
|---|---|---|---|---|
| MOL-RAO cloud radar | −18.4 [−22.0 to −15.5] | −24.4 [−30.5 to −17.2] | −13.9 [−22.7 to −7.3] | −40.1 [−51.2 to −30.4] |
| ICON-LEM control (C2R) | −16.2 [−20.5 to −11.7] | −15.6 [−22.3 to −5.2] | −4.9 [−7.0 to −2.3] | −8.1 [−13.5 to −0.9] |
| ICON-LEM perturbed (P2R) | −17.8 [−21.5 to −13.4] | −21.5 [−27.4 to −16.9] | −9.9 [−15.1 to −5.5] | −14.4 [−21.6 to −8.2] |

Revised :

**Table 5.** Median as well as $25^{\text{th}}$ and $75^{\text{th}}$ percentiles of the cloud radar reflectivity (Ze) distributions on 2 May 2013 for the four classes described in Fig. 7.

|  | Drizzle onset Mean [$25^{\text{th}}$ to $75^{\text{th}}$] | Drizzle growth Mean [$25^{\text{th}}$ to $75^{\text{th}}$] | Drizzle mature Mean [$25^{\text{th}}$ to $75^{\text{th}}$] | Precipitation Mean [$25^{\text{th}}$ to $75^{\text{th}}$] |
|---|---|---|---|---|
| MOL-RAO cloud radar | −18.4 [−22.1 to −15.4] | −24.2 [−30.3 to −17.0] | −14.2 [−23.1 to −7.5] | −38.2 [−49.7 to −28.2] |
| ICON-LEM control (C2R) | −16.2 [−20.5 to −11.7] | −15.6 [−22.3 to −5.2] | −4.9 [−7.0 to −2.3] | −8.1 [−13.5 to −0.9] |
| ICON-LEM perturbed (P2R) | −17.8 [−21.5 to −13.4] | −21.5 [−27.4 to −16.9] | −9.9 [−15.1 to −5.5] | −14.4 [−21.6 to −8.2] |

**Page 16, line 14-18:**

Original: The comparison between cloud reflectivities (and Doppler velocities and spectrum width) to the corresponding forward modeled variables from ICON-LEM simulations shows a fair agreement between observations and model. However, despite the effort to make model and data comparable, and despite the rather strong signal in the model, this is not yet a useful tool for detection and attribution of differences between control (C2R) and perturbed simulations (P2R) in this case. This tool is at a preliminary stage and can be used to evaluate microphysical schemes to observations (Acquistapace, 2017).

Revised: In conclusion, despite the effort to make model and data comparable using the forward operator, and despite the rather strong signal in the model, no detection and attribution of an aerosol signal could be achieved. In the future more comparisons are needed as difference at one grid point only could arise from a different sampling of cloud life cycle.

**Page 16, line 22:**

Original: […] are compared with high-resolution ceilometer measurements (15 s temporal resolution) from the DWD ceilometer network.

Revised: […] are compared with high-resolution ceilometer measurements (15 s temporal resolution) from the DWD ceilometer network (please see Section 2.3.2 for details).

**Page 16, line 24:**

Sentence added: The problem is not due to issues with the initial- or boundary conditions, as the discrepancy to the reference observations is larger in the outer nests (312 and 624 m resolution, respectively; result not shown here).

**Page 17, Figure 7:**
Original:

[Figure]

Revised:

[Figure]

**Page 17, Figure 7 caption:**
Original: […] as in Acquistapace et al. (2017) [...]
Revised: […] as in Acquistapace et al. (2019) [...]

**Page 19, line 16:**
Sentence added: A part of the difference in simulated and observed cloud types can be attributed to a difference in the spatial distribution of the cloud types at the MODIS overpass time.

**Page 19, line 18:**
Original: Figure 10 displays PDFs of Nd, [...]
Revised: Figure 10 displays normalized frequency of occurrence distributions of Nd, [...]

**Page 20, line 29:**
Original: In order to put this number into context, we assessed the aerosol ERF from four different models from the 5th Coupled 30 Model Intercomparison Project (CMIP5; Taylor et al., 2012) for which the relevant output diagnostics were available to infer the time series of the aerosol ERF. This time series of the ERF over the industrial period was computed by Kretzschmar et al. (2017). The average aerosol ERF (both, aerosol-cloud and aerosol-radiation interactions; the latter are not considered in our current modelling study) for the annual average 1850 to 2000 (present-day minus pre-industrial) as global annual mean was -1.2 W m−2. The aerosol ERF over central Europe (the domain investigated here) in May, averaged over 1983 to 1987, was -4.0 W m−2. This latter is larger than the global-annual average present-day vs. pre-industrial, since it is over a region with a large local aerosol perturbation which is even much larger than present-day minus pre-industrial, and the solar zenith angle in May is larger than in the annual mean. In total, a scaling factor of 3.4 is obtained. The ICON-LEM thus implies a global annual mean ERF due to aerosol-cloud interactions in 2000 of about -0.8 W m−2.

Revised: In order to put this number into context, we assess the ERF computed by general circulation models. Within the 6$^{th}$ Coupled Model Intercomparison Project (CMIP6; Eyring et al., 2016), the Radiative Forcing Model Intercomparison Project (RFMIP; Pincus et al., 2016) defined a simulation dedicated to the assessment of the transient historical effective radiative forcing, the simulation "RFMIP-ERF-HistAerO3". There are four models in the CMIP6 archive that submitted output for these simulations, namely the CanESM5 (Swart et al., 2019), the GFDL-CM4 (Held et al., 2019), the MIROC6 (Tatebe et al., 2019), and the NorESM2-LM (Bentsen et al., 2013; Kirkevåg et al., 2018), which supplied 3, 1, 3, and 2 ensemble members, respectively. The ERFaci is approximated by using the change in cloud radiative effect (CRE, the difference between all-sky and clear-sky top-of-atmosphere net radiation flux density here taken in the solar spectrum only) between two time periods (Quaas et al., 2009). When evaluating the difference in solar CRE of the individual years 2013 and 1850, one obtains as the difference in global annual mean, a multi-model mean of -0.81 Wm$^{-2}$, with an inter-model standard deviation of 0.34 Wm$^{-2}$ (using all ensemble members). For a five-year average difference(2010 to 2014 and 1850 to 1854; the periods are not centered around the specific years because the simulations run from 1850 to 2014; Pincus et al., 2016), the values for the global annual mean are -0.71 ± 0.35 W m$^{-2}$. For the domain of the ICON-LEM simulation, and only using May (only monthly output is available), the signal, defined as the difference 1985 minus 2013 is much more noisy since it is averaged much less. The mean and standard deviation are -4.69±13.05 Wm$^{-2}$. To assess the uncertainty, we computed also the change in solar CRE for the months April and May (since the actual day is early May), averaged over the five year-periods 1983 to 1987 minus 2010 to 2014, and a larger domain (10°W to 30°E, 40° to 60°N). This yields a smaller value and much smaller standard deviation of -2.55±2.99 Wm$^{-2}$. The scaling factor for the five-year and bigger European domain is 3.6; the one for the single years and ICON-LEM domain is 5.8. The uncertainty in these scaling factors obtained from the GCMs is very large. Nevertheless it may be instructive to know that the forcing for May, considering the large difference in aerosol levels over Europe between 1985 and 2013 is a factor of 4 to 6 larger than considering the global ERFaci between 2013 and 1850. The -2.6 W m$^{-2}$ obtained in this simulation (Table 7) thus would imply a global, annual mean ERFaci for 2013 vs. 1850 of between -0.4 and -0.7 Wm-2.

**Page 21, Fig. 10 caption:**
Original: Figure 10. Probability distributions of […]. The numbers of the right side show the number of pixels used for the PDF calculation.
Revised: Figure 10. Normalized frequency of occurrence distributions of […]. The numbers of the right side show the number of pixels used for the normalized frequency of occurrence distribution calculation.

**Page 22, line 26:**
Sentence added: Although the simulations in this study are limited to one day over the domain of Germany, this work shows the great potential of combining these new high resolution simulations with a large set of observations for the detection and attribution of aerosol-cloud interactions. In the future this work should be complemented by extended analyses for longer time periods and more regions to further improve our understanding of cloud-aerosol interactions.

**Page 22, line 31:**

[revised manuscript text omitted]